# Nonsense-mediated mRNA decay factor UPF1 promotes aggresome formation

Yeonkyoung Park [1,2,5], Joori Park [1,2,5], Hyun Jung Hwang [1,2,5], Byungju Kim [3,5], Kwon Jeong[1,2], Jeeyoon Chang [1,2], Jong-Bong Lee [3,4✉] & Yoon Ki Kim [1,2✉]

Nonsense-mediated mRNA decay (NMD) typifies an mRNA surveillance pathway. Because NMD necessitates a translation event to recognize a premature termination codon on mRNAs, truncated misfolded polypeptides (NMD-polypeptides) could potentially be generated from NMD substrates as byproducts. Here, we show that when the ubiquitin–proteasome system is overwhelmed, various misfolded polypeptides including NMD-polypeptides accumulate in the aggresome: a perinuclear nonmembranous compartment eventually cleared by autophagy. Hyperphosphorylation of the key NMD factor UPF1 is required for selective targeting of the misfolded polypeptide aggregates toward the aggresome via the CTIF–eEF1A1–DCTN1 complex: the aggresome-targeting cellular machinery. Visualization at a single-particle level reveals that UPF1 increases the frequency and fidelity of movement of CTIF aggregates toward the aggresome. Furthermore, the apoptosis induced by proteotoxic stresses is suppressed by UPF1 hyperphosphorylation. Altogether, our data provide evidence that UPF1 functions in the regulation of a protein surveillance as well as an mRNA quality control.

[1] Creative Research Initiatives Center for Molecular Biology of Translation, Korea University, Seoul 02841, Republic of Korea. [2] School of Life Sciences, Korea University, Seoul 02841, Republic of Korea. [3] Department of Physics, Pohang University of Science and Technology (POSTECH), Pohang 37673, Republic of Korea. [4] School of Interdisciplinary Bioscience and Bioengineering, POSTECH, Pohang 37673, Republic of Korea. [5] These authors contributed equally: Yeonkyoung Park, Joori Park, Hyun Jung Hwang, Byungju Kim. ✉email: jblee@postech.ac.kr; yk-kim@korea.ac.kr

Upstream frameshift 1 (UPF1), a monomeric ATP-dependent RNA helicase belonging to superfamily 1, is a core factor of nonsense-mediated mRNA decay (NMD), the best-characterized mRNA quality control pathway in eukaryotic cells[1–4]. With UPF1 dependency as a hallmark, NMD plays a dual role in eukaryotic gene expression as an mRNA surveillance pathway [by rapidly eliminating faulty transcripts harboring a premature termination codon (PTC)] and as a post-transcriptional gene-regulatory pathway (by destabilizing normally generated transcripts).

The NMD pathway is mechanistically subdivided into two steps: substrate recognition and substrate degradation[3,5]. Selective substrate recognition takes place at the translation termination step largely during the pioneer (or first) round of translation of newly synthesized mRNAs with the 5′ cap bound to the nuclear cap-binding complex: a heterodimer of cap-binding proteins (CBPs) 20 and 80[6–8]. In accordance with conventional NMD, when newly synthesized mRNAs contain the exon junction complex (EJC) sufficiently downstream of the termination codon, they are recognized as NMD substrates. The cellular factors recruited to the termination codon—eukaryotic peptide chain release factors 1 and 3, UPF1, and SMG1 (a kinase specific for UPF1)—form a SURF complex[9]. Then, an interaction between the SURF complex and a downstream EJC leads to activation of UPF1, i.e., hyperphosphorylation of UPF1 by SMG1. Consequently, the termination codon is recognized as a PTC. After that, the hyperphosphorylated UPF1 triggers rapid degradation of this mRNA either endoribonucleolytically or exoribonucleolytically by recruiting decay-inducing factors, such as SMG5, SMG6, SMG7, and/or PNRC2[3,5].

Just as mRNAs are subject to quality control, the quality of polypeptides is also monitored by surveillance pathways in mammalian cells[10–15]. Misfolded polypeptides can be generated during gene expression in various ways: genetic mutations causing amino acid alterations, an aberrant translation event that generates defective ribosomal products, anomalous post-translational modifications, and inefficient ribosome quality control. In addition, truncated (and possibly misfolded) nascent polypeptides could be generated as byproducts during NMD because NMD necessitates a translation event to recognize a PTC on an mRNA[11,12].

The misfolded polypeptides are prone to aggregation and consequently elicit proteotoxic stresses. To cope with misfolded-polypeptide–induced proteotoxicity, eukaryotic cells have evolved several mechanisms[10–15]. Molecular chaperones assist with refolding of misfolded polypeptides into their correct three-dimensional structure. Alternatively, the ubiquitin–proteasome system (UPS) disposes of the misfolded polypeptides. Nonetheless, when the refolding mediated by molecular chaperones and degradation by the UPS are impaired or overwhelmed, potentially toxic misfolded polypeptides tend to form small cytoplasmic aggregates[16–18]. Then, these aggregates are selectively recognized by aggresome-targeting cellular factors or machineries[19,20], such as histone deacetylase 6 (HDAC6), the BAG3–chaperon complex, and/or a CED complex composed of nuclear cap-binding complex–dependent translation initiation factor (CTIF), eukaryotic translation elongation factor 1 alpha 1 (eEF1A1), and dynactin 1 (DCTN1). After selective recognition, the small cytoplasmic aggregates are transported to the microtubule-organizing center (MTOC) via microtubule-mediated retrograde movement, thus forming a perinuclear structure called the aggresome[16–18]. The misfolded polypeptides accumulated in the aggresome are eventually cleared from the cell via the autophagy pathway: this type of autophagy is named aggrephagy[20]. Notably, many neurodegenerative diseases are linked to intracellular inclusion bodies (biochemically and morphologically similar to the aggresome observed in cultured cells) such as intracellular Lewy bodies containing aggregated α-synuclein[21–23].

In this study, we provide molecular evidence for a role of a key NMD factor, UPF1, in protein quality control. According to our results, a variety of misfolded polypeptides including truncated polypeptides potentially generated from PTC-containing NMD substrates (NMD-polypeptides) are moved into the aggresome via CED-mediated retrograde transport. In particular, such an aggresomal targeting of misfolded polypeptides depends on UPF1 hyperphosphorylation. Furthermore, a single-particle experiment reveals that UPF1 increases the frequency of movement of CTIF aggregates toward the aggresome and ensures proper movement of these aggregates to the aggresome. Accordingly, the apoptosis induced by proteotoxic stresses is also dependent on UPF1 and its hyperphosphorylation.

## Results

**NMD-polypeptides accumulate in the aggresome.** Because NMD is a translation-coupled mRNA quality control pathway[3,5], the PTC-containing NMD substrates generate truncated polypeptides, which could be misfolded and deleterious to normal cell functions. It is also known that NMD largely occurs during the pioneer round of translation[6] and that CTIF—a specific factor involved in the pioneer round of translation[24]—forms the CED complex (along with eEF1A1 and DCTN1), which recognizes and transports misfolded polypeptide aggregates to the aggresome[19,25]. On the basis of the above observations, we hypothesized that NMD-polypeptides are transported to the aggresome via the CED complex.

To test this hypothesis, we first designed an NMD reporter mRNA: a FLAG-tagged glutathione peroxidase 1 (*GPx1*) mRNA harboring either a normal termination codon (Norm) or PTC (Ter; Fig. 1a). As expected, the reporter was efficiently downregulated by NMD, as evidenced by an increase in the relative abundance of FLAG-GPx1-Ter mRNA after downregulation of already known NMD factors (UPF1, UPF2, and UPF3B), EJC core components (eIF4A3 and Y14), or a specific factor involved in the pioneer round of translation (CTIF; Supplementary Fig. 1).

After the NMD reporter validation, we assessed intracellular distribution of the FLAG-GPx1-Norm or -Ter polypeptides in HeLa cells stably expressing green fluorescent protein (GFP)-fused cystic fibrosis transmembrane conductance regulator ΔF508 (CFTR-ΔF508), which contains a single amino acid deletion at position 508 of the CFTR protein and forms an aggresome in cultured cells[17,25,26]. The results of conventional confocal microscopy revealed that FLAG-GPx1-Norm polypeptides were evenly distributed throughout the cell regardless of treatment with MG132, a potent proteasome inhibitor (Fig. 1b). In contrast, the FLAG-GPx1-Ter polypeptides were hardly detectable under normal conditions. Notably, MG132 treatment caused significant enrichment of the FLAG-GPx1-Ter polypeptides in a cytoplasmic inclusion (Fig. 1b), which overlapped with the distribution of a known aggresomal protein, CFTR-ΔF508, implying that FLAG-GPx1-Ter polypeptides are targeted to the aggresome. Further analysis showed that the cytoplasmic inclusion body of the FLAG-GPx1-Ter polypeptides overlapped with γ-tubulin (a component of MTOC, where the aggresome forms[18]; Fig. 1c), CTIF and DCTN1 (CED components[25]; Supplementary Fig. 2a), and the puromycin-conjugated polypeptide (polypeptidyl-puro; Supplementary Fig. 2a). Puromycin treatment results in premature translation termination and generates polypeptidyl-puro[27], which is a truncated and potentially misfolded defective ribosomal product. Polypeptidyl-puro was found to be transported to the

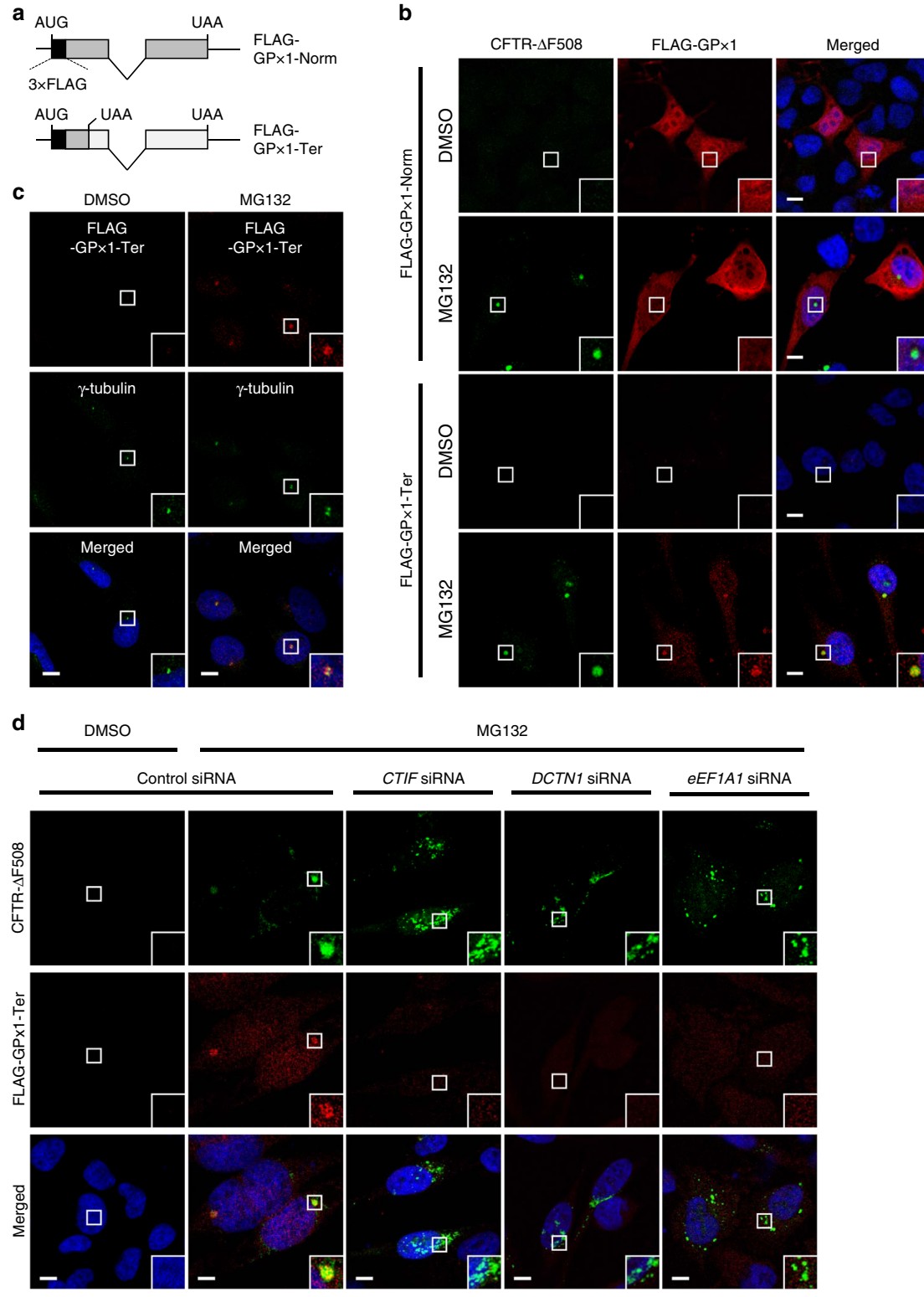

aggresome at a lower concentration of puromycin or to both the aggresome and aggresome-like induced structures at a higher concentration of puromycin (Supplementary Fig. 2b). Moreover, downregulation of a CED component (CTIF, DCTN1, or eEF1A1) by means of small interfering RNA (siRNA) caused dispersion of the aggresome containing the FLAG-GPx1-Ter polypeptides (Fig. 1d). Therefore, all our data indicate that truncated and potentially misfolded NMD-polypeptides are targeted to the aggresome.

## Aggresomal targeting of misfolded polypeptides requires UPF1

We next investigated how the aggresomal targeting of NMD-polypeptides is triggered during NMD. Given that NMD-polypeptides are generated in the course of translation of PTC-containing mRNAs, it is likely that the PTC recognition step and/or downstream steps during NMD may be coupled to aggresomal targeting of NMD-polypeptides. To test this possibility, we assessed the relative change in the distribution of FLAG-GPx1-Ter polypeptides in HeLa cells stably expressing CFTR-ΔF508

**Fig. 1 NMD-polypeptides are targeted to the aggresome. a** The schematic diagram of NMD reporter mRNAs: FLAG-GPx1-Norm and -Ter [harboring either a normal termination codon or premature termination codon (PTC), respectively]. The regions encoding three tandem repeats of the FLAG tag (3xFLAG) and *GPx1* are specified by black and dark gray boxes, respectively. AUG, translation initiation codon; UAA, translation termination codon. **b** Immunostaining of CFTR-ΔF508 (green) and either FLAG-GPx1-Norm or -Ter polypeptides (red). HeLa cells stably expressing CFTR-ΔF508 were transiently transfected with a plasmid expressing either 3xFLAG-GPx1-Norm or -Ter. The cells were treated with either dimethyl sulfoxide (DMSO) or MG132 for 12 h before immunostaining. Nuclei were stained with 4′,6-diamidino-2-phenylindole (DAPI; blue). A magnified view of the white boxed area is provided in the lower right corner of each image. **c** Immunostaining of a known aggresomal marker (γ-tubulin; green) and the FLAG-GPx1-Ter polypeptides (red). **d** Immunostaining of CFTR-ΔF508 (green) and the FLAG-GPx1-Ter polypeptides (red) after downregulation of a component of the CED complex (CTIF, eEF1A1, or DCTN1). HeLa cells stably expressing CFTR-ΔF508 were transfected with the indicated siRNA. Two days later, the cells were retransfected with a plasmid expressing FLAG-GPx1-Ter. The cells were treated with either DMSO or MG132 for 12 h before the immunostaining. Immunostaining images in each panel are representative of at least two independently processed biological replicates (*n* = 3 for (**b**) and (**c**) and *n* = 2 for (**d**). Scale bar, 10 μm.

after specific downregulation of an NMD factor. Downregulation of an NMD factor (UPF1) or a CED component (DCTN1, which served as a positive control), but not downregulation of other tested NMD factors (PNRC2, SMG5, SMG6, or SMG7), caused almost complete dispersion of the aggresome containing NMD-polypeptides (Fig. 2a). In addition, downregulation of either UPF1 or DCTN1 triggered the dispersion of the aggresome containing CFTR-ΔF508 (Fig. 2a, b) or polypeptidyl-puro (Fig. 2c; Supplementary Fig. 3a). Specific downregulation by each siRNA was confirmed by western blotting (Supplementary Fig. 3b). Of note, downregulation of a CED component (eEF1A1 or DCTN1) did not significantly affect NMD efficiency (Supplementary Fig. 3c,d). Therefore, all these findings indicate that the key NMD factor UPF1 is required for selective targeting of various types of misfolded polypeptides (such as NMD-polypeptides, CFTR-ΔF508, and polypeptidyl-puro) to the aggresome.

**Hyperphosphorylated UPF1 is targeted to the aggresome**. It is well known that efficient NMD requires continuous alternation of phosphorylation and dephosphorylation of UPF1[3,5]. Therefore, we investigated possible involvement of UPF1 phosphorylation in the aggresome formation. To this end, we employed *UPF1* siRNA-resistant (R) Myc-tagged wild type (WT) UPF1 and its three variants: hyperphosphorylated (HP), HP-12A, and HP-E3 mut. UPF1-HP contains two substitutions (G495R and G497E) within the helicase domain, and as a consequence, it lacks ATPase activity and becomes locked on mRNAs, leading to its hyperphosphorylation[28–31]. UPF1-HP-12A contains two amino acid substitutions (G495R and G497E) for hyperphosphorylation and 12 amino acid substitutions (from serine or threonine to alanine) at the positions experimentally proved to be phosphorylated by SMG1[31], and therefore, is expected to not be phosphorylated even though it contains HP-inducing substitutions. UPF1-HP-E3 mut contains two amino acid substitutions (G495R and G497E) for hyperphosphorylation and three additional amino acid substitutions (S124A, N138A, and T139A) for disrupting a putative E2-binding pocket and accordingly is hyperphosphorylated and lacks its E3 ubiquitin ligase activity[32].

We first confirmed the phosphorylation status of UPF1-WT and its variants. For this purpose, we carried out immunoprecipitation (IP) experiments in RNase A–treated extracts of the cells expressing UPF1-WT or its variants followed by western blotting with an anti (α)-phospho (p)-(S/T)Q antibody (Fig. 3a and Supplementary Fig. 4a). The results showed that UPF1-WT, most of which is known to be hypophosphorylated[28,33,34], was phosphorylated at a basal level. In contrast, UPF1-HP and UPF1-HP-E3 mut were phosphorylated strongly. On the other hand, UPF1-HP-12A manifested a weak but significant level of the phosphorylation, suggesting that additional residues other than the previously characterized 12 residues[31] could be phosphorylated on the UPF1-HP backbone. It should also be noted that

NMD inhibition by UPF1 downregulation was almost completely reversed by the expression of UPF1-WT, but not other UPF1 variants (Supplementary Fig. 4b, c).

We next monitored the intracellular distributions of UPF1 and its variants (Fig. 3b). In line with previous observations[35,36], under normal conditions, UPF1-WT, UPF1-HP-12A, and UPF1-HP-E3 mut were evenly distributed throughout the cytoplasm or slightly enriched in processing bodies: cytoplasmic nonmembranous compartments composed of translationally repressed messenger ribonucleoproteins (mRNPs) and 5′-to-3′ mRNA decay factors[37] [Fig. 3b, dimethyl sulfoxide (DMSO) treatment]. When the cells were treated with MG132, the distributions of UPF1-WT and UPF1-HP-12A did not change significantly (Fig. 3b, MG132 treatment). In contrast, UPF1-HP almost completely overlapped with the CFTR-ΔF508 aggresome. UPF1-HP-E3 mut was also found to be enriched in the aggresome, suggesting that aggresomal targeting of UPF1 is dependent on its hyperphosphorylation but not relevant to its E3 ligase activity. Of note, the expression of UPF1-HP-12A caused a disruption of the CFTR-ΔF508 aggresome, suggesting that UPF1-HP-12A functions dominant-negatively in aggresome formation. Additional data from conventional confocal microscopy showed that the intracellular distribution of NMD factors (PNRC2, SMG5, and SMG6), a component of the decapping complex (DCP1A), or a nuclear cap-binding complex component (CBP80) did not overlap with the CFTR-ΔF508 aggresome during MG132 treatment (Supplementary Fig. 5), suggesting that all the tested NMD-related factors except UPF1 are not relevant to aggresomal targeting. Collectively, all these results indicate that UPF1 is transported to the aggresome in a UPF1 phosphorylation-dependent manner. In agreement with this conclusion, treatment of the cells with okadaic acid (a potent inhibitor of protein phosphatases 2A), which causes accumulation of hyperphosphorylated UPF1, triggered aggresomal targeting of Myc-UPF1-WT (Supplementary Fig. 6).

**Aggresome formation requires UPF1 hyperphosphorylation**. Considering that (i) UPF1 is required for the formation of an aggresome containing misfolded polypeptides (Fig. 2) and (ii) hyperphosphorylated UPF1 accumulates in the aggresome (Fig. 3), it is likely that UPF1 hyperphosphorylation is needed for efficient formation of the aggresome. To test this possibility, we conducted complementation experiments with *UPF1* siRNA, siRNA-resistant Myc-UPF1-WT or one of its variants (Fig. 3a), and MG132-treated HeLa cells stably expressing CFTR-ΔF508. Selective downregulation of endogenous UPF1 as well as protein expression of Myc-UPF1-WT and its variants at the approximate level of endogenous UPF1 were confirmed by western blotting (Fig. 4a). The UPF1 downregulation triggered substantial dispersion of the CFTR-ΔF508 aggresome (Fig. 4b, c). The dispersion pattern was partially or remarkably reversed when

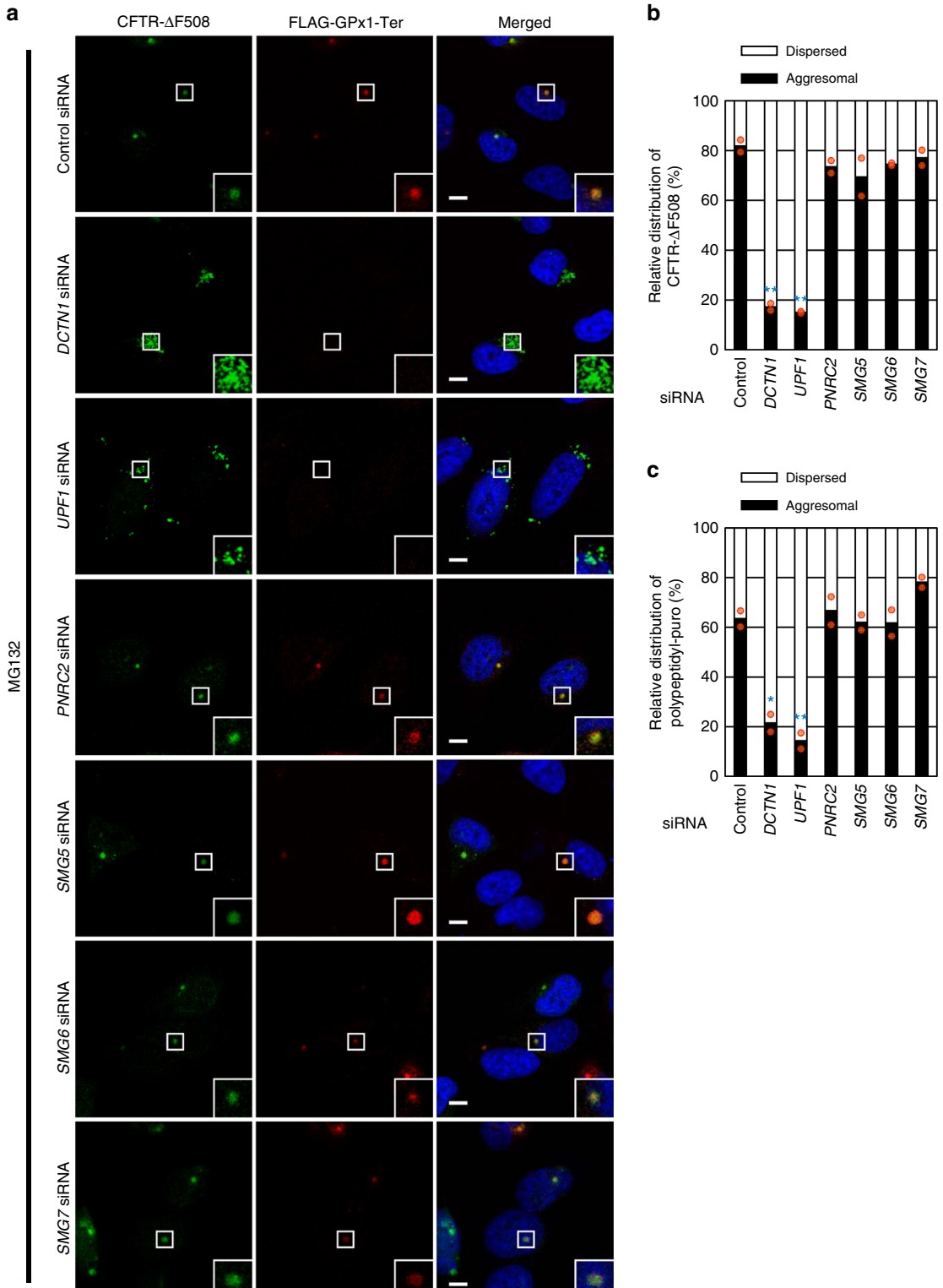

**Fig. 2 UPF1 is involved in the formation of aggresome. a** Immunostaining of CFTR-ΔF508 (green) and the FLAG-GPx1-Ter polypeptides (red) after downregulation of NMD factors. HeLa cells stably expressing CFTR-ΔF508 were transfected with the indicated siRNAs. Two days later, the cells were retransfected with the plasmid expressing FLAG-GPx1-Ter. The cells were treated with MG132 for 12 h before the immunostaining. Scale bar, 10 μm; $n = 2$. **b** Relative percentages of the cells containing either the aggresome or dispersed aggregates of CFTR-ΔF508. Relative percentages of the cells were determined by counting of the cells containing either the aggresome or dispersed aggregates of CFTR-ΔF508 in the immunostaining images. Two-tailed, equal-sample variance Student's $t$ test was carried out to calculate the $P$ values. **$P = 0.0019$ (*DCTN1* siRNA) and 0.0014 (*UPF1* siRNA); More than 100 cells were analyzed from each of two independent experiments. **c** Relative percentages of the cells containing either the aggresome or dispersed aggregates of polypeptidyl-puro. As performed in panel b, except that immunostaining images from Supplementary Fig. 3a were analyzed. Two-tailed, equal-sample variance Student's $t$ test was carried out to calculate the $P$ values. *$P = 0.0127$ (*DCTN1* siRNA), **$P = 0.0085$ (*UPF1* siRNA); More than 100 cells were analyzed from each of two independent experiments. Source data are provided as a Source Data File.

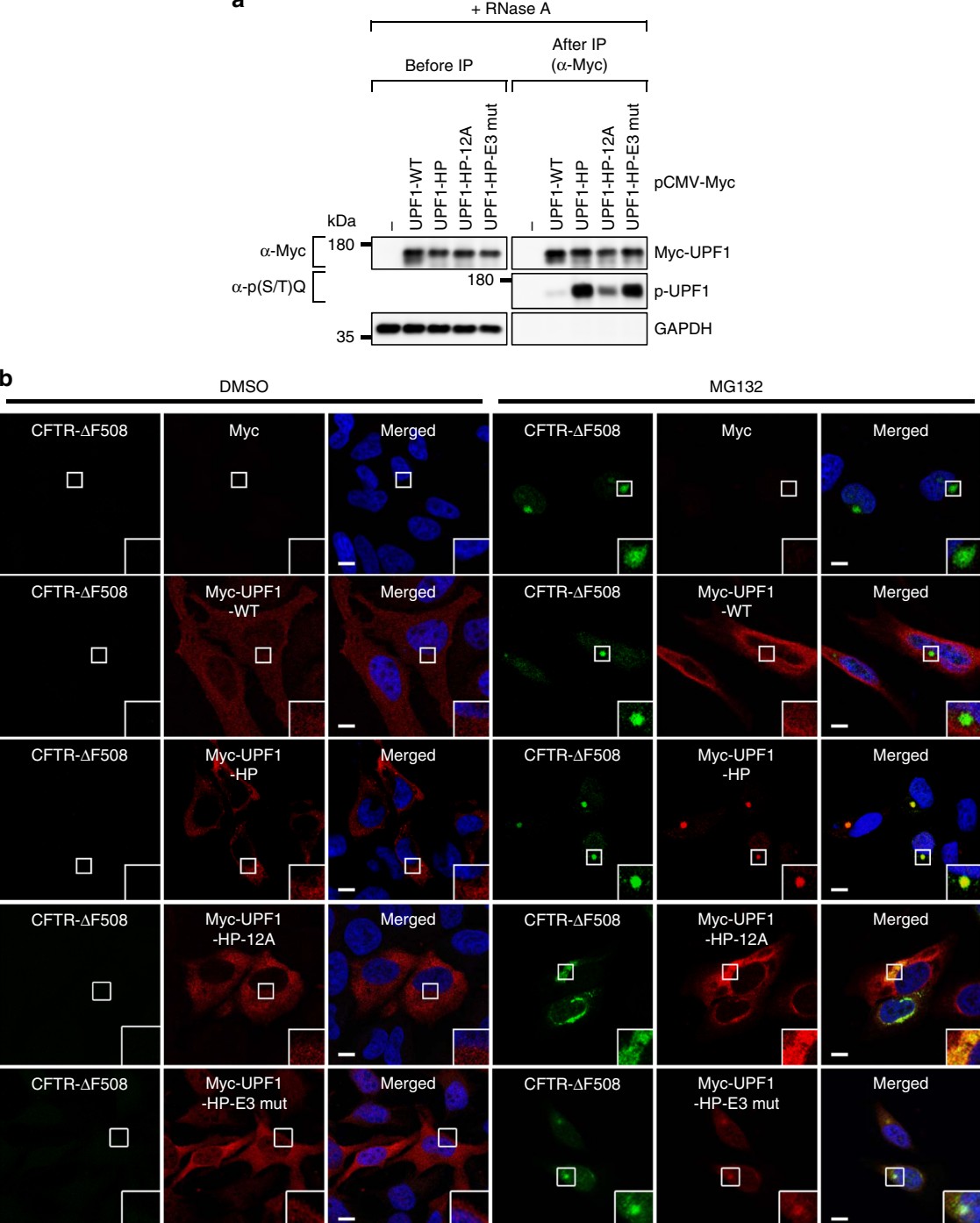

**Fig. 3 Localization of UPF1 to the aggresome depends on its phosphorylation status. a** Validation of UPF1 hyperphosphorylation. HEK293T cells were transiently transfected with a plasmid expressing either Myc-UPF1-WT (wild type) or one of its variants. Two days later, the cells were harvested. Next, the extract of cells was treated with RNase A before IPs and was subjected to IPs using the anti (α)-Myc antibody. The samples before and after IPs were analyzed by western blotting; $n = 2$. Source data are provided as a Source Data File. **b** Immunostaining of CFTR-ΔF508 (green) and either Myc-UPF1-WT or its variants (red). HeLa cells stably expressing CFTR-ΔF508 were transiently transfected with the indicated plasmid expressing either Myc-UPF1-WT or one of its variants. Immunostaining images are representative of three independently processed biological replicates. Scale bar, 10 μm; $n = 3$.

Myc-UPF1-WT or Myc-UPF1-HP was expressed, respectively. On the other hand, the expression of Myc-UPF1-HP-12A failed to reverse the observed dispersion of CFTR-ΔF508 (Fig. 4c, compare with the aggresomal fraction). Of note, two-thirds of dispersed CFTR-ΔF508 in the cells expressing Myc-UPF1-HP-12A yielded a distribution overlapping with that of Myc-UPF1-HP-12A (Fig. 4c, overlapped and dispersed fraction), suggesting

that Myc-UPF1-HP-12A has an ability to associate with misfolded polypeptides but fails to form an aggresome containing misfolded polypeptides. Overall, these data imply that efficient aggresome formation involves UPF1 hyperphosphorylation. In support of this conclusion, downregulation of a known UPF1 kinase[3,5] (SMG1 or ATM), but not DNA-PKcs, triggered the dispersion of the CFTR-ΔF508 aggresome (Supplementary

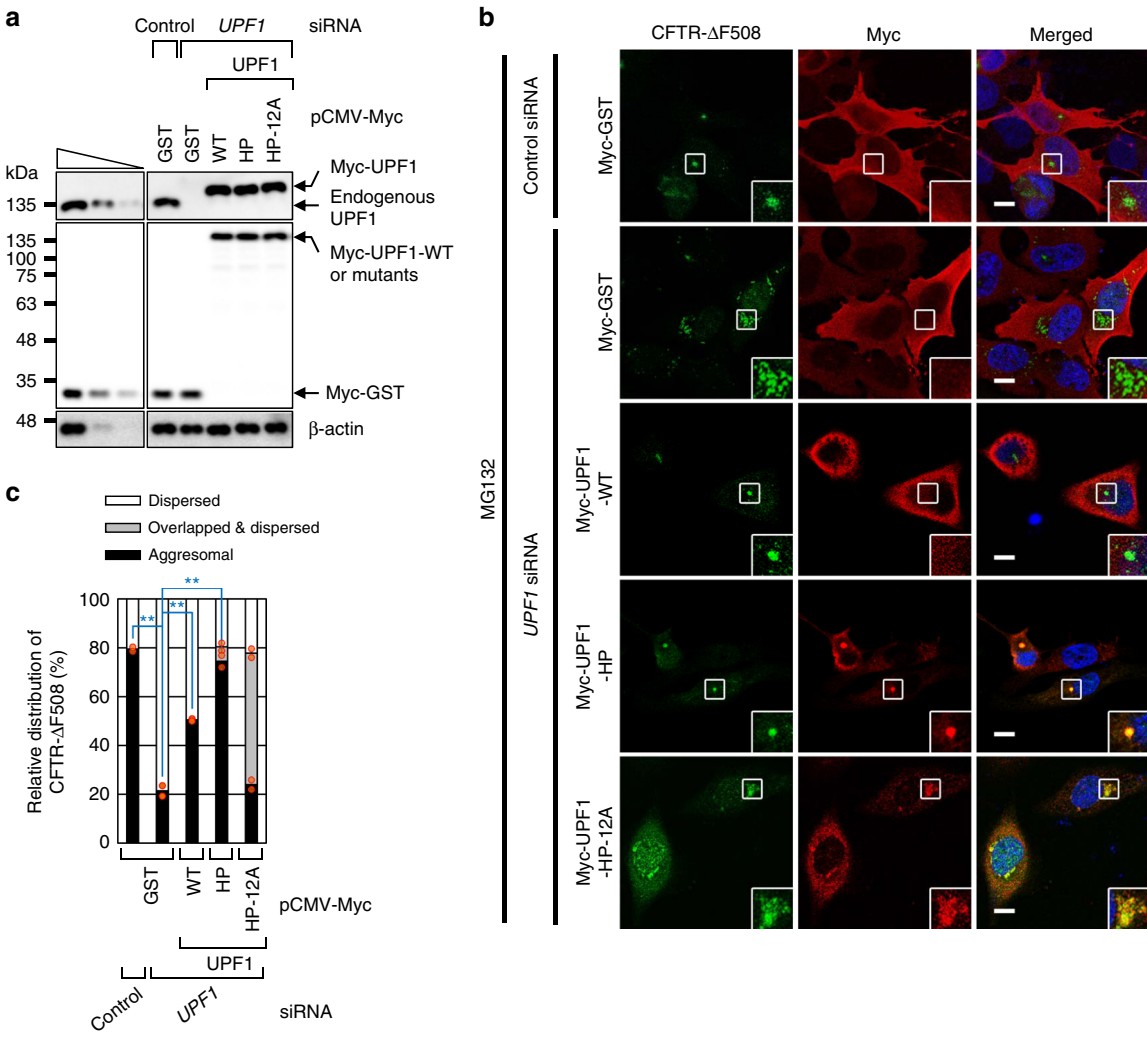

**Fig. 4 Efficient aggresome formation requires UPF1 hyperphosphorylation.** HeLa cells stably expressing CFTR-ΔF508 were either depleted or not depleted of endogenous UPF1. One day later, the cells were retransfected with a plasmid expressing siRNA-resistant UPF1 [either wild type (WT) or one of the variants] or a plasmid expressing GST, which served as a negative control. Then, the influence of UPF1 on CFTR-ΔF508 aggresome formation was analyzed by immunostaining. **a** Western blots showing selective downregulation of UPF1 and comparable expression levels of Myc-UPF1 and endogenous UPF1; $n = 2$. **b** Immunostaining of CFTR-ΔF508 (green) and either exogenous UPF1 or GST (red). Scale bar, 10 μm; $n = 2$. **c** Relative percentages of the cells with different intracellular distributions of CFTR-ΔF508: aggresomal, dispersed, or dispersed but overlapping with exogenous UPF1. As performed in Fig. 2, except that immunostaining images from panel b were analyzed. To accurately assess the effect of exogenously expressed UPF1 on aggresome formation, the distributions of CFTR-ΔF508 were determined only in the cells expressing either exogenous UPF1 or GST. Two-tailed, equal-sample variance Student's $t$ test was carried out to calculate the $P$ values. $**P < 0.0057$. The exact $P$ values are provided in a Source Data File; More than 100 cells were analyzed from two independent experiments; Source data are provided as a Source Data File.

Fig. 7a–c). We also observed that overexpression of CFTR-ΔF508 triggers phosphorylation of UPF1 in a way that depends on SMG1 and ATM (Supplementary Fig. 7d).

Considering that UPF1-WT almost completely restored the NMD efficiency inhibited by UPF1 downregulation in this study (Supplementary Fig. 4b, c), the observed partial restoration of aggresome formation by Myc-UPF1-WT (Fig. 4c) may be due to a local change in structure (important for aggresome formation) within UPF1 after N-terminal Myc tagging. Alternatively, exogenously expressed UPF1 may require additional time for the full ability to trigger efficient aggresome formation [e.g., its association with a misfolded polypeptide, its uncharacterized protein folding, formation of its complex with CED, or its movement (toward an aggresome) in the form of a complex with the misfolded polypeptide].

**HP-UPF1 more strongly associates with the CED complex**. To understand the molecular mechanism underlying the participation of UPF1 in aggresome formation, we conducted IPs using FLAG-UPF1-WT or its variants in RNase A–treated extracts of the cells. The western blotting of immunoprecipitated FLAG-UPF1 with an α-p-(S/T)Q antibody revealed that ~8-fold more UPF1-HP was hyperphosphorylated, as compared with UPF1-WT or UPF1-HP-12A (Fig. 5a, b). Consistent with the results of other studies[28,35,36], under normal conditions, approximately ~5-fold more DCP1A coimmunopurified with UPF1-HP than with either UPF1-WT or UPF1-HP-12A. In the same IPs, we observed that 4- to 6-fold greater amounts of CED components (CTIF, eEF1A1, and DCTN1) coimmunopurified with UPF1-HP than with UPF1-WT or UPF1-HP-12A, suggesting the preferential association between hyperphosphorylated UPF1 and the CED

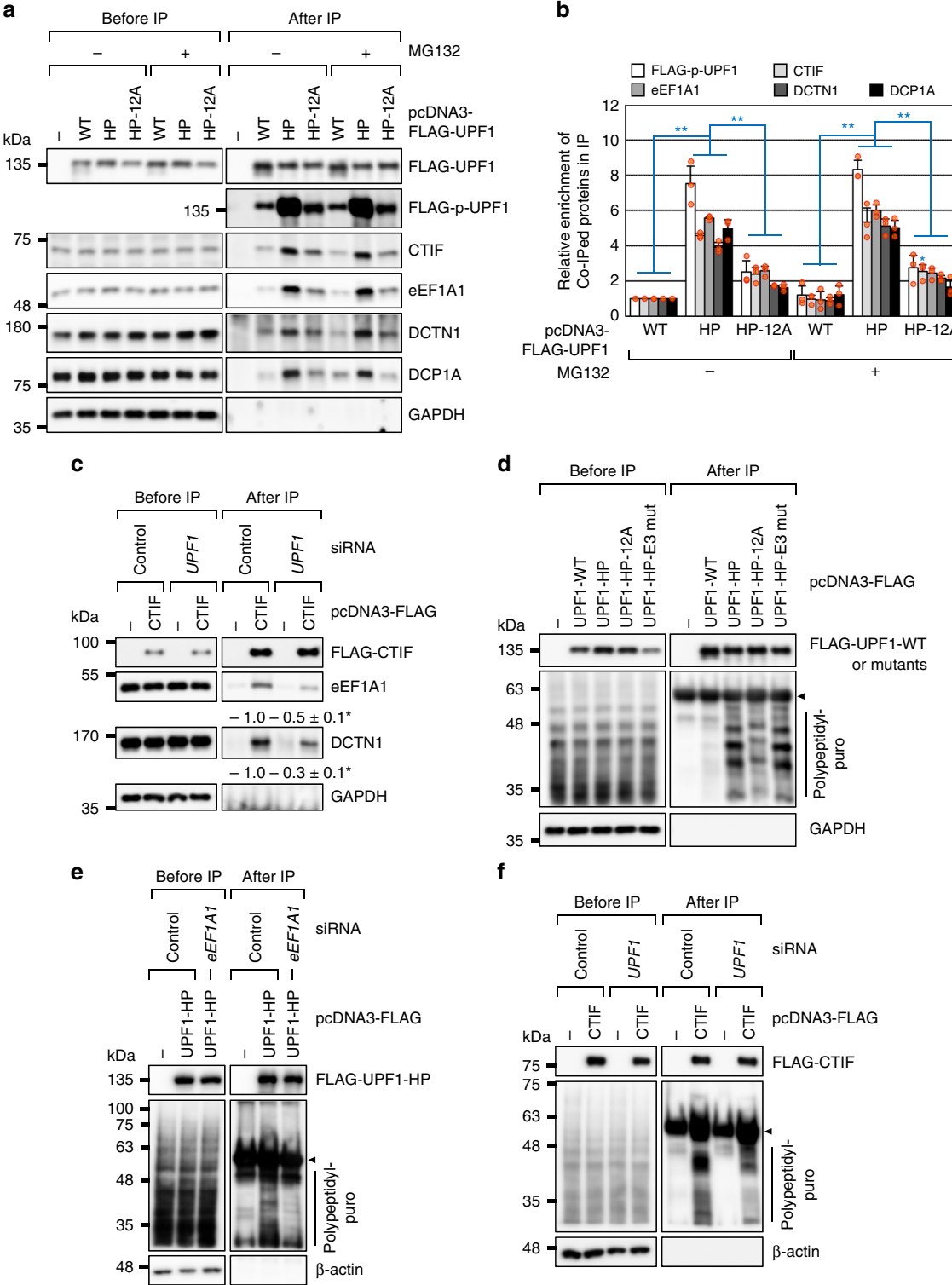

complex. Notably, all the observed cases of preferential enrichment in IPs of UPF1-HP were not significantly affected by treatment with MG132. Reciprocal IPs involving FLAG-CTIF also supported the preferential association between UPF1-HP and the CED complex regardless of MG132 treatment (Supplementary Fig. 8a). In agreement with the results of another study showing dissociation of the CED complex from translating mRNPs when the cells were treated with MG132[25], the treatment with MG132 disrupted the association between CTIF and specific cellular factors (CBP80 and eIF3b) associated with mRNPs undergoing

the pioneer round of translation (Supplementary Fig. 8a). Furthermore, other IP experiments using the extracts of the cells treated with okadaic acid pointed to the increased association between hyperphosphorylated UPF1 and the CED complex (Supplementary Fig. 8b). It should be noted that a possible RNA-mediated protein–protein association could be ruled out under our IP conditions. This was evident from a basal level of endogenous *GAPDH* mRNA in RNase A–treated extracts (Supplementary Fig. 8c) and a reduced amount of coimmunopurified CBP80 (which is known to interact with UPF1 in a partially

**Fig. 5 Hyperphosphorylated UPF1 promotes CED complex formation. a** IPs of FLAG-UPF1: either WT or one of its variants. HEK293T cells transiently expressing the indicated protein were treated with either DMSO or MG132 for 12 h before harvesting. The cell extracts were treated with RNase A and then subjected to IPs with the α-FLAG antibody; n = 3. **b** Quantitation of the Western blotting images presented in (**a**). Signal intensities of the Western blot bands were quantified. The levels of hyperphosphorylated UPF1 and coimmunopurified cellular proteins were normalized to the amounts of immunopurified FLAG-UPF1. The normalized levels obtained in the IP of FLAG-UPF1-WT were arbitrarily set to 1.0. Data are presented as mean values ± standard deviations (SD) and statistical significance; Two-tailed, equal-sample variance Student's t test was carried out to calculate the P values. *P = 0.0102 and **P < 0.0039. The exact P values are provided in a Source Data File; n = 3. **c** IPs of FLAG-CTIF in cells depleted of UPF1. HEK293T cells either depleted or not depleted of endogenous UPF1 were transiently transfected with a plasmid expressing FLAG-CTIF. The cells were treated with MG132 for 12 h before harvesting. The extracts of the cells were treated with RNase A and then subjected to IPs using the α-FLAG antibody; Two-tailed, equal-sample variance Student's t test was carried out to calculate the P values. *P = 0.0241 (eEF1A1) and 0.0184 (DCTN1); n = 3. **d** IPs of FLAG-UPF1-WT or its variants in the extracts of cells treated with puromycin. HEK293T cells transiently expressing FLAG-UPF1-WT or one of its variants were treated with MG132 for 12 h and puromycin for 1 h before cell harvesting. The RNase A–treated extracts of the cells were subjected to IPs with the α-FLAG antibody. The immunoglobulin heavy chain detected after IPs is denoted by an arrowhead; n = 2. **e** IPs of FLAG-UPF1-HP using the extracts of cells depleted of eEF1A1. As performed in panel d, except that HEK293T cells either depleted or not depleted of endogenous eEF1A1 were transiently transfected with a plasmid expressing FLAG-UPF1-HP; n = 2. **f** IPs of FLAG-CTIF in the extracts of cells depleted of UPF1. As performed in panel d, except that HEK293T cells either depleted or not depleted of endogenous UPF1 were transiently transfected with a plasmid expressing FLAG-CTIF; n = 2; Source data are provided as a Source Data File.

RNA-dependent manner[29,38,39]) in the IP of UPF1 (Supplementary Fig. 8d). Collectively, all these results indicate that UPF1 is complexed with the CED complex in a phosphorylation-dependent and MG132 treatment–independent manner.

**UPF1 promotes the formation of the CED complex**. On the basis of the preferential association between UPF1-HP and the CED complex (Fig. 5a, b; Supplementary Fig. 8), we next investigated the molecular role of UPF1 in CED complex formation. First, in vitro GST pull-down analysis showed that recombinant UPF1 preferentially interacted with recombinant GST-CTIF but not with GST, GST-eEF1A1, or GST-DCTN1 (Supplementary Fig. 9a). Second, the IP experiments with FLAG-CTIF-WT or FLAG-CTIF(54-598), which lacks an N-terminal region responsible for the binding to DCTN1 and eEF1A1[25], implied that UPF1 preferentially associated with FLAG-CTIF-WT, in contrast to FLAG-CTIF(54-598) (Supplementary Fig. 9b). Third, the IP experiments using FLAG-CTIF or FLAG-eEF1A1 and the RNase A–treated extracts of cells revealed that ~2-fold less DCTN1 and eEF1A1 and ~2-fold less DCTN1 and CTIF coimmunopurified with FLAG-CTIF (Fig. 5c) and FLAG-eEF1A1 (Supplementary Fig. 9c), respectively, when UPF1 was downregulated. Specific downregulation of UPF1 was demonstrated by western blotting (Supplementary Fig. 9d). Taken together, these findings indicate that UPF1 helps to maintain the integrity of the CED complex via its preferential binding to the N-terminal region of CTIF.

**HP-UPF1 is complexed with misfolded polypeptides**. In the CED complex, eEF1A1 is responsible for selective recognition of both pre-existing and newly synthesized misfolded polypeptides[40–42]. Therefore, considering our observations that UPF1-HP strongly associates with the CED complex (Fig. 5a, b; Supplementary Fig. 8a), we next tested whether UPF1 can bind to misfolded polypeptides via the CED complex. To this end, we carried out IP experiments with either FLAG-UPF1-WT or one of its variants and the RNase A–treated extracts of the cells transiently treated with both puromycin and MG132. The IP data showed that a basal and small amount of newly synthesized (and potentially misfolded) polypeptidyl-puro coimmunopurified with UPF1-WT and UPF1-HP-12A (which was confirmed to be minimally phosphorylated; Fig. 3a), respectively (Fig. 5d). The IP data showed that, in agreement with our finding that UPF1-HP-12A is still phosphorylated to some extent as compared with UPF1-WT (Fig. 3a), slightly more amount of newly synthesized (and potentially misfolded) polypeptidyl-puro coimmunopurified with UPF1-HP-12A compared with UPF1-WT (Fig. 5d). On the

other hand, a greater amount of polypeptidyl-puro coimmunopurified with UPF1-HP and UPF1-HP-E3 mut, in contrast to UPF1-WT and UPF1-HP-12A. Of note, the association between polypeptidyl-puro and UPF1-HP was inhibited by eEF1A1 downregulation (Fig. 5e; Supplementary Fig. 9e). Moreover, in line with our observation that UPF1 helps to maintain the integrity of the CED complex (Fig. 5c; Supplementary Fig. 9c), the association between polypeptidyl-puro and CTIF was suppressed by UPF1 downregulation (Fig. 5f; Supplementary Fig. 9f). Thus, the above results indicate that the association between hyperphosphorylated UPF1 and misfolded polypeptides depends on the CED complex.

**UPF1 increases the frequency of aggresomal movement**. To directly analyze UPF1-dependent aggresome formation from CTIF-bound aggregates, we visualized the motion of individual GFP-CTIF aggregates on microtubules in live mouse embryonic fibroblasts (MEFs) by line-scan confocal microscopy[43]. Visualization of microtubules by treatment of cells with silicon rhodamine (SiR)-conjugated tubulin, which enables specific labeling of microtubules in live cells, proved that the GFP-CTIF aggregates migrated toward the aggresome along microtubules (Fig. 6a; Supplementary Movie 1). The GFP-CTIF aggregates moved toward the aggresome at a translocation rate of $2.6 \pm 0.8 \, \mu m \, s^{-1}$ (mean ± SD) and $2.7 \pm 0.9 \, \mu m \, s^{-1}$ under normal conditions and during MG132 treatment, respectively (Supplementary Fig. 10a; Supplementary Movie 2), suggesting that the aggresome-targeting machinery itself (CED complex) is constitutively active rather than induced by the accumulation of misfolded polypeptides. Furthermore, in agreement with another report, which suggests that the CED complex transports misfolded polypeptides to the aggresome along microtubules via its association with dynein motor proteins[25], the measured translocation rate of GFP-CTIF aggregates was comparable to that of a dynein motor protein on a microtubule in vivo $(2.4 \pm 0.8 \, \mu m \, s^{-1})$[44]. Of note, the translocation rate of GFP-CTIF moving away from an aggresome was also comparable to that toward the aggresome (Supplementary Fig. 10b). These results suggest that the movement of CTIF away from the aggresome—this step is necessary for efficient recycling of CTIF—is an active process rather than simple diffusion.

We next investigated the function of UPF1 in aggresomal targeting by measuring the number and rate of GFP-CTIF aggregates migrating toward the aggresome at the single-particle level, when MEFs were either depleted of mouse (m) UPF1 or not (Fig. 6b–e; Supplementary Fig. 11; Supplementary Movie 3). Specific downregulation was confirmed by western blotting (Supplementary Fig. 11a). To determine the direction and

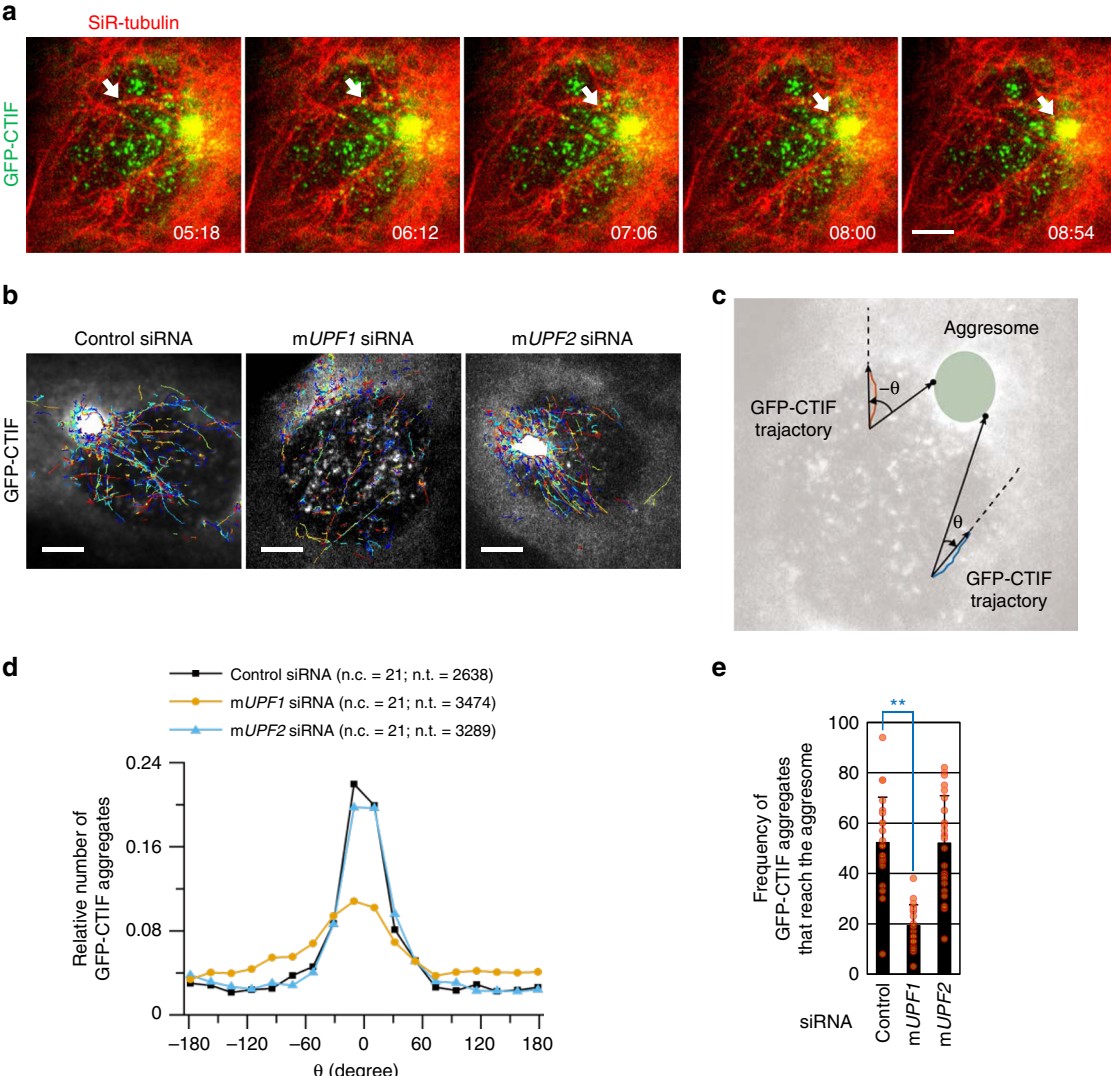

**Fig. 6 UPF1 increases the frequency and fidelity of movement of CTIF aggregates. a** Live cell imaging of GFP-CTIF (green) with SiR-conjugated tubulin (red). MEFs were transiently transfected with GFP-CTIF and were treated with SiR-tubulin for staining of microtubules. Then, the cells were visualized by line-scan confocal microscopy. Scale bar, 5 μm; *n* = 3. **b** GFP-CTIF aggregates moving toward the aggresome after downregulation of mouse (m) UPF1 or mUPF2. MEFs were transfected with m*UPF1* siRNA, m*UPF2* siRNA, or nonspecific control siRNA. One day later, the cells were transiently transfected with GFP-CTIF. Two days later, the cells were treated with MG132 and visualized by line-scan confocal microscopy. The trajectories of individual GFP-CTIF aggregates that moved over 590 nm are depicted. Scale bar, 5 μm; *n* = 3. **c** The degree of the migration direction of individual GFP-CTIF aggregates with respect to the aggresome. We first determined the brightest position corresponding to the aggresome periphery closest to the GFP-CTIF aggregates of interest. Then, we measured the angle between the linear trajectory of the GFP-CTIF and a line connecting the brightest position with the initial point of the trajectory of GFP-CTIF. **d** The distribution of the observed angle values of individual GFP-CTIF aggregates. The number of the cells at each angle was normalized to the total number of the cells analyzed. The number of cells (n.c.) were 21 from three independent experiments for all cases and the number of trajectories (n.t.) were 2638 (Control siRNA), 3474 (m*UPF1* siRNA), and 3289 (m*UPF2* siRNA), respectively. **e** The frequency of GFP-CTIF aggregates that reach the aggresome. The number of GFP-CTIF aggregates (per cell) that reached the aggresome during 15 s were counted. Data are presented as mean values ± SD. Two-tailed, unequal-sample variance Student's *t* test was carried out to calculate the *P* values. \*\**P* < 0.000. The exact *P* values are provided in a Source Data File; *n* = 23 (Control siRNA), 22 (m*UPF1* siRNA), and 25 (m*UPF2* siRNA) cells from three independent experiments were examined; Source data are provided as a Source Data File.

number of migrating GFP-CTIF aggregates at the same time, we defined the degree of directional motion of GFP-CTIF aggregates toward the aggresome as an angle (θ) between the direction of movement of the GFP-CTIF aggregates and the location of the aggresome periphery (Fig. 6c). The results revealed that the distribution of the angle values was drastically spread out in the mUPF1-depleted cells, in comparison with undepleted or mUPF2-depleted cells (Fig. 6d). Furthermore, downregulation of mUPF1, but not mUPF2, significantly decreased the frequency of movement of GFP-CTIF aggregates toward the aggresome

(Fig. 6e), without affecting the translocation rate (Supplementary Fig. 11b). Of note, neither mUPF1 nor mUPF2 downregulation affected the organization of microtubule and MTOC, whereas treatment with the microtubule-depolymerizing drug nocodazole almost completely disrupted microtubule organization without affecting MTOC (Supplementary Fig. 12). Thus, taken together, these results suggest that UPF1 promotes the formation of aggresome containing misfolded polypeptides at two distinct steps: it facilitates efficient formation of the CED complex containing misfolded polypeptides (Fig. 5) and ensures proper

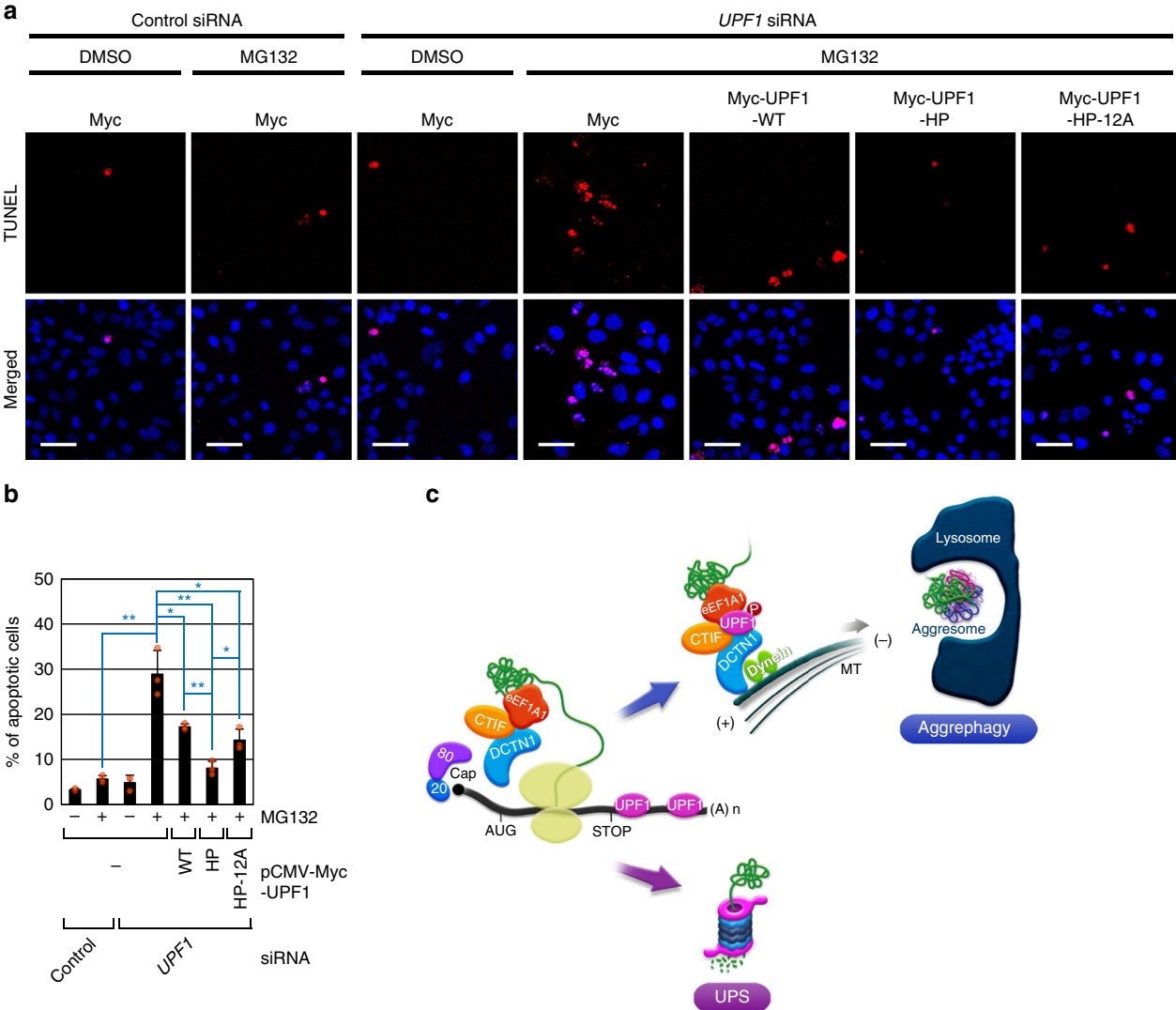

**Fig. 7 UPF1 hyperphosphorylation represses apoptosis induced by proteotoxic stresses. a**, **b** The influence of UPF1 hyperphosphorylation on the apoptosis induced by proteotoxic stresses. Stably CFTR-ΔF508–expressing HeLa cells either depleted or not depleted of endogenous UPF1 were retransfected with a plasmid expressing Myc-UPF1-WT or one of its variants. One day after transfection, the cells were treated with either DMSO or MG132 for 16 h. **a** The population of apoptotic cells was visualized by TUNEL assay. Scale bar, 50 μm; n = 3. **b** TUNEL-positive cells were quantitated, and the relative percentage of apoptotic cells is presented. Data are presented as mean values ± SD. Two-tailed, equal-sample variance Student's *t* test was carried out to calculate the P values. *P < 0.0208; **P < 0.0027. The exact P values are provided in a Source Data File; n = 3; Source data are provided as a Source Data File. **c** The proposed model of the role of UPF1 in aggresome formation.

movement of the misfolded-polypeptide–associating CED complex to the aggresome (Fig. 6).

**HP-UPF1 renders cells resistant to proteotoxic stresses.** Misfolded polypeptides accumulating in the aggresome are eventually degraded by the autophagy pathway[20]. Otherwise, the small cytoplasmic aggregates containing misfolded polypeptides would cause proteotoxic stresses thereby inducing apoptosis[25,26]; such aggregates are associated with various neurodegenerative diseases[16–18]. Because hyperphosphorylation of UPF1 is required for aggresome formation (Figs. 3, 4), we investigated possible participation of UPF1 hyperphosphorylation in the apoptosis induced by proteotoxic stresses. For this purpose, we employed HeLa cells stably expressing a misfolded polypeptide, CFTR-ΔF508, and quantitated the apoptotic cell populations after MG132 treatment in a terminal deoxynucleotidyl transferase dUTP nick end labeling (TUNEL) assay.

In agreement with the results of other studies[25,26], the treatment of undepleted HeLa cells (stably expressing CFTR-ΔF508) with MG132 led to minimal apoptosis (~5.6% of all cells; Fig. 7a, b). In contrast, the treatment of the UPF1-depleted cells with MG132 significantly increased the apoptotic population (to ~29% of all cells). This increase in the apoptotic population was partially or markedly reversed by the expression of UPF1-WT or UPF1-HP, respectively, implying that UPF1 hyperphosphorylation suppresses the apoptosis induced by proteotoxic stresses. Furthermore, the expression of UPF1-HP-12A partially reversed the increase in the apoptotic population. Considering our observations that UPF1-HP-12A is weakly but more phosphorylated than UPF1-WT is (Fig. 3a), it is likely that the observed partial restoration of the apoptosis by UPF1-HP-12A may be due to the partial phosphorylation of UPF1-HP-12A. Alternatively, given that (i) UPF1-HP-12A associates with both CED components and misfolded polypeptides more strongly than UPF1-WT does (Fig. 5a–d) and (ii) although the expression of UPF1-HP-12A

fails to restore efficient formation of an aggresome, its intracellular distribution overlaps with that of dispersed CFTR-ΔF508 aggregates (Fig. 4), the association between UPF1-HP-12A and misfolded polypeptides, and therefore, their colocalization may partially relieve proteotoxic stress. Of note, individual down-regulation of other NMD factors did not significantly affect the proportion of apoptotic cells (Supplementary Fig. 13a–c), suggesting that the observed reversal of apoptosis by the expression of UPF1-HP is not related to NMD inhibition. In support of this notion, in the same condition, NMD efficiency of the reporter mRNAs was significantly inhibited by UPF1 downregulation (Supplementary Fig. 13d). This inhibition was remarkably attenuated by complementation with UPF1-WT, but not UPF1-HP or UPF1-HP-12A. Collectively, all these findings indicate that hyperphosphorylated UPF1 suppresses the apoptosis induced by proteotoxic stresses.

## Discussion

To maintain high fidelity of gene expression, eukaryotic cells have evolved multilayer post-transcriptional and post-translational quality control pathways[10–15]. In this study, we unravel a previously unappreciated role of UPF1 in the specific targeting of misfolded polypeptides to the aggresome, where the misfolded polypeptides will be eventually eliminated from the cell by the aggresome–autophagy pathway[16–18,20].

When newly synthesized mRNAs containing a PTC(s) are subjected to the pioneer round of translation, they are recognized by NMD machinery (Fig. 7c). After the selective substrate recognition, UPF1 activated (hyperphosphorylated) by SMG1 leads to rapid degradation of NMD substrates. At the same time, the potentially misfolded NMD-polypeptides synthesized during the pioneer round of translation are rapidly degraded by the UPS. During this process, the E3 ubiquitin ligase activity of UPF1 may participate in the UPS-mediated degradation of NMD-polypeptides, as suggested by various studies[32,45–48]. Nonetheless, when the UPS is overwhelmed or NMD-polypeptides are not properly ubiquitinated, the newly synthesized NMD-polypeptides are subject to a fail-safe surveillance pathway: they are selectively recognized by eEF1A1 within a CED complex. The association between the CED complex and NMD-polypeptides is promoted by the UPF1 hyperphosphorylated during NMD. Then, the CED complex containing NMD-polypeptides moves toward the aggresome via microtubule-mediated retrograde transport. In addition, proper guidance of the CED complex containing NMD-polypeptides toward the aggresome is ensured by hyperphosphorylated UPF1. The NMD-polypeptides accumulated in the aggresome are eventually eliminated by the autophagy pathway. In this way, UPF1 orchestrates and ensures the correctness of both mRNAs and polypeptides at the same time.

It should be noted that the involvement of UPF1 in aggresome formation is not limited to NMD-polypeptides. We observed that aggresomal targeting of various misfolded polypeptides such as CFTR-ΔF508 and polypeptidyl-puro also requires UPF1 (Figs. 2–4). In the case of NMD-polypeptides, it is likely that UPF1 hyperphosphorylated by SMG1 during NMD is preferentially engaged in the formation of the aggresome containing NMD-polypeptides. In contrast, in the case of non–NMD-polypeptides, the emergence of misfolded polypeptides synthesized from non-NMD substrates can induce proteotoxic stress, activating UPF1 kinases. Because UPF1 promiscuously binds to mRNA with enrichment in the 3′ untranslated region[49], the UPF1 in close proximity of misfolded polypeptides being generated from the mRNA may be preferentially phosphorylated by UPF1 kinases. Indeed, we observed that overexpression of a misfolded polypeptide triggers UPF1 phosphorylation via a stress-induced

kinase, ATM (Supplementary Fig. 7d). In addition, down-regulation of ATM caused the dispersion of the aggresome containing CFTR-ΔF508 (Supplementary Fig. 7a–c). Therefore, as long as UPF1 is hyperphosphorylated regardless of the method of its hyperphosphorylation, the activated UPF1 can promote efficient formation of the aggresome containing misfolded polypeptides.

In addition to NMD, several other RNA surveillance pathways coupled to a translation event can generate aberrant and potentially misfolded polypeptides[11–15]. In particular, recent studies showed that nascent truncated polypeptides synthesized from mRNPs undergoing ribosome stalling are subject to cotranslational quality control. This process is mediated by a ribosome quality control complex composed of listerin (a ribosome-associated E3 ubiquitin ligase), a nuclear-export mediator, and other accessory proteins, and consequently, leads to ubiquitination and rapid degradation of the aberrant nascent polypeptides via the UPS[50,51]. Nevertheless, when nascent truncated polypeptides synthesized by a stalled ribosome are not properly ubiquitinated or not efficiently degraded because of a UPS impairment, they can be aggregated and transported toward the aggresome[52–54]. Considering that hyperphosphorylated UPF1 can direct the movement of misfolded polypeptides (generated from either NMD substrates or non-NMD substrates) toward the aggresome in a CED-dependent manner, it is plausible that hyperphosphorylated UPF1 is engaged in the formation of the aggresome containing nascent truncated polypeptides synthesized by the stalled ribosome.

In our study, we demonstrated the cytoprotective effect of UPF1 against proteotoxic stresses induced by accumulation of misfolded polypeptides (Fig. 7). In line with our observations, several recent studies implicated the alleviation of proteotoxicity by UPF1 in neurodegenerative diseases. For instance, aggregate-prone proteins—transactive response element DNA/RNA-binding protein of 43 kDa (TDP43) and fused in sarcoma (FUS)—are enriched in inclusion bodies within the neuronal cells of patients with amyotrophic lateral sclerosis (ALS) and frontotemporal dementia (FTD)[55]. Such aggregation or inclusion of either TDP43 or FUS generates proteotoxic stresses that are linked to the development of ALS and FTD[55]. In a primary neuronal model of ALS and FTD or in yeast, UPF1 has been found to protect the cell from the proteotoxic stresses induced by the aggregation of either TDP43 or FUS[56–58]. Therefore, considering that (i) CTIF is enriched in the inclusion bodies seen in neurodegenerative diseases[25], (ii) UPF1 associates with the CED complex (Fig. 5), and (iii) UPF1 is required for efficient CED-mediated targeting of misfolded polypeptides toward the aggresome (Figs. 2–4), our data provide molecular insights into a cytoprotective function of UPF1 in neurodegenerative diseases linked with accumulation of misfolded polypeptides.

## Methods

**Cell culture and chemical treatment**. HeLa and HEK293T cells (purchased from ATCC), and MEFs (a gift from Byung-Yoon Ahn, Korea University, Seoul, Korea) were maintained in Dulbecco's modified Eagle's medium (DMEM; Capricorn Scientific) containing 10% fetal bovine serum (Capricorn Scientific) and 1% penicillin/streptomycin (Capricorn Scientific). HeLa cells stably expressing CFTR-ΔF508[25] were cultured in the normal growth media containing 0.4 mg ml$^{-1}$ G418.

To induce the formation of an aggresome containing misfolded polypeptides, cells were treated with either MG132 (5 μM; Calbiochem) or, as a negative control, DMSO (BioShop) for 12 h (for IP and immunostaining) or for 16 h (for the TUNEL assay). Where indicated, the cells were also treated with puromycin (10 μg ml$^{-1}$ for IPs or 1 μg ml$^{-1}$ for immunostaining) for 1 h, okadaic acid (30 nM) for 12 h, or nocodazole (1 μM) for 12 h before cell harvesting. For line-scan confocal microscopy, the cells were treated with MG132 for < 2 h before visualization.

Plasmocin (Invivogen) was employed to minimize mycoplasma contamination, and the MycoAlert PLUS Mycoplasma Detection Kit (Lonza) was used for mycoplasma detection.

**Plasmid construction**. The following plasmids have been described previously: p3xFLAG-CMV™-7.1 (renamed to p3xFLAG in this study; Sigma); pCMV-Myc (Clontech); pRSET A (Invitrogen); pcDNA3-FLAG, pcDNA3-FLAG-CTIF-WT, pCMV-Myc-CTIF, pEGFP-C1-CTIF, pmCMV-GPx1-Norm or -Ter, pGEX-6p-1, and phCMV-MUP[24]; p3xFLAG-eEF1A1, pCMV-Myc-GST, and pCMV-Myc-DCTN1[25]; pCMV-Myc-UPF1-WT and pCMV-Myc-UPF1(G495R/G497E) renamed to pCMV-Myc-UPF1-HP in this study[28]; pCMV-Myc-PNRC2, pcDNA3-FLAG-UPF1-WT, and pcDNA3-FLAG-UPF1-(G495R/G497E) renamed to pcDNA3-FLAG-UPF1-HP in this study[35]; pcDNA3-FLAG-SMG5 and pcDNA3-FLAG-SMG6[59]; pCMV-Myc-DCP1A[60]; GFP-CFTR-ΔF508[17]; and pcDNA3.1-CBP80-HA[61].

To construct p3xFLAG-GPx1-Norm or -Ter, the GPx1-Norm or -Ter fragment was amplified by PCR using pmCMV-GPx1-Norm or -Ter as a template and two oligonucleotides: 5′-CATGCCATGGCCATGTCTGCTGCTCGGCTCTCCGCGG TGG-3′ (sense) and 5′-GCTCTAGATTAGGGGTTGCTAGGCTGCTTGGACAG C-3′ (antisense). The PCR-amplified and Klenow-filled NcoI/XbaI fragment was ligated into a Klenow-filled HindIII/XbaI fragment of p3xFLAG.

To construct pCMV-Myc-UPF1-HP-12A, the 12 serine or threonine residues (positions 10, 28, 1038, 1041, 1046, 1050, 1055, 1073, 1078, 1089, 1096, and 1116) experimentally proved to be phosphorylated by SMG1[31] were replaced by alanines in pCMV-Myc-UPF1-HP.

To construct pCMV-Myc-UPF1-HP-E3 mut, the fragment of UPF1 cDNA harboring substitutions S124A/N138A/T139A was synthesized in vitro. Then, the corresponding region in pCMV-Myc-UPF1-HP was replaced by the in vitro–synthesized UPF1 fragment.

To generate pcDNA3-FLAG-UPF1-HP-12A, a Klenow-filled HindIII/NotI fragment of pCMV-Myc-UPF1-HP-12A was ligated into a Klenow-filled Acc65I fragment of pcDNA3-FLAG.

To construct pGEX-6p-1-CTIF, a BamHI/EcoRI fragment of pSK(-)-CTIF[24] was ligated to a BamHI/EcoRI fragment of pGEX-6p-1.

To construct pGEX-6p-1-eEF1A1, a Klenow-filled BamHI/SalI fragment of p3x-FLAG-eEF1A1[25] was ligated into a Klenow-filled NotI/SalI fragment of pGEX-6p-1.

To construct pGEX-6p-1-DCTN1, an EcoRI/BspEI fragment containing the 5′ half of DCTN1 cDNA was amplified by PCR with pCMV-Myc-DCTN1 as a template and two oligonucleotides: 5′-CCGGAATTCATGGCACAGAGCAAGA GGCACGTGTAC-3′ (sense) and 5′-TGGCCTGCAGCAGGCTCAGCGAGTAC-3′ (antisense). pCMV-Myc-DCTN1 was digested with BspEI and NotI to obtain the 3′ half of DCTN1 cDNA. The resulting two fragments were ligated to an EcoRI/NotI fragment of pGEX-6p-1.

To construct pRSET A-UPF1, a Klenow-filled BamHI/EcoRI fragment of pCMV-Myc-UPF1-WT was ligated to a Klenow-filled HindIII/EcoRI fragment of pRSET A.

**DNA or siRNA transfection**. For IP experiments or conventional confocal microscopy, HeLa cells or HEK 293T cells were transfected with a plasmid using calcium phosphate or Lipofectamine 2000 (Invitrogen). MEFs were transfected by means of Lipofectamine 3000 (Invitrogen). For specific downregulation of endogenous proteins, cells were transfected with 100 nM in vitro–synthesized siRNAs via Oligofectamine (Life Technologies) or Lipofectamine 3000 (Invitrogen). The siRNA sequences used in this study are listed in Supplementary Data 1.

**Quantitative reverse-transcription PCR (qRT-PCR)**. Total-RNA samples were purified with the TRIzol Reagent (Life Technologies) and analyzed by qRT-PCRs. The total-RNA samples were converted into complementary DNA (cDNA) using RevertAid Reverse Transcriptase (Thermo Scientific). qRT-PCR analyses were carried out with gene-specific oligonucleotides and the Light Cycler 480 SYBR Green I Master Mix (Roche) on a Light Cycler 480 II machine (Roche). The following gene-specific oligonucleotides for amplification of mRNAs were used in this study: 5′-GGACTACAAAGACCATGACG-3′ (sense) and 5′-CTTCTCAC-CATTCACCTCGCACTT-3′ (antisense) for amplification of FLAG-GPx1-Norm or -Ter mRNAs; 5′-TGGCAAATTCCATGGCACC-3′ (sense) and 5′-AGA-GATGATGACCCTTTTG-3′ (antisense) for amplification of GAPDH mRNAs; 5′-CAACACCCCAACATCTTCG-3′ (sense) and 5′-CTTTCCGCCCTTCTTGGCC-3′ (antisense) for amplification of FLuc RNAs; and 5′-CTGATGGGGGCTCTATG-3′ (sense) and 5′-TCCTGGTGAGAAGTCTCC-3′ (antisense) for amplification of MUP mRNAs.

To validate efficient removal of cellular RNAs after RNase A treatment under our conditions (Supplementary Figs. 4a and 8c), total-RNA samples purified from the extracts either treated or not treated with RNase A before IPs were mixed with equal amounts of in vitro–synthesized FLuc RNAs (10 pg) as a spike-in to adjust the data for differences among RNA preparations. Then, the relative amounts of endogenous GAPDH mRNA (normalized to FLuc RNA) were compared. Microsoft Excel (Microsoft) was used for analyzing of quantitative real-time RT-PCR data.

**Antibodies**. Antibodies against the following proteins (used for immunostaining, IPs, and western blotting) have been described previously: CTIF and CBP80[24]; eIF4A3, UPF1, and UPF2[62]; UPF3B and SMG6[63]; phospho-S1078-UPF1 and phospho-S1096-UPF1[62]; and PNRC2[35].

Antibodies against the following proteins were purchased [listed in the format "protein name (catalog number, supplier, dilution fold)"]: FLAG (DYKDDDDK; 14793, Cell Signaling Technology, 1:50, or A8592, Sigma, 1:5000), HA (3724, Cell Signaling Technology, 1:200), Myc (9E10; OP10L, Calbiochem, 1:5000, or 2272, Cell Signaling Technology, 1:200), GFP (sc-9996, Santa Cruz Biotechnology, 1:50), DCTN1 (p150^glued; 610474, BD Biosciences, 1:250), eIF1A1 (CBP-KK1; EF1α; 05-235, Merck Millipore, 1:1000), DCP1A (D5444, Sigma, 1:1000), Y14 (RBM8A; MAB2484, Abnova, 1:1000), SMG1 (A300-394A, Bethyl Laboratories, 1:1000), SMG5 (ab33033, Abcam, 1:500), SMG7 (A302-170A, Bethyl Laboratories, 1:2000), ATM (A300-299A, Bethyl Laboratories, 1:1000), DNA-PKcs (A300-517A, Bethyl Laboratories, 1:1000), p-(S/T)Q ATM/ATR substrate (2851, Cell Signaling Technology, 1:1000), puromycin (12D10; MABE343, Merck Millipore, 1:10000 for western blotting or 1:400 for immunostaining), eIF3b (sc-16377, Santa Cruz Biotechnology, 1:1000), γ-tubulin (sc-17788, Santa Cruz Biotechnology, 1:20), α-tubulin (sc-53030, Santa Cruz Biotechnology, 1:100), GST (A190-122A, Bethyl Laboratories, 1:8000), His (27-4710-01, GE Healthcare, 1:3000), β-actin (A5441, Sigma, 1:10,000), and GAPDH (LF-PA0212, AbFrontier, 1:10,000).

**Immunostaining followed by conventional confocal microscopy**. Immunostaining was performed on HeLa cells as described elsewhere[25]. The cells were fixed and permeabilized with 3.65–3.8% formaldehyde (Sigma, cat. # F8775) and 0.5% Triton X-100 (Sigma), respectively. Then, the fixed cells were incubated with 1.5% bovine serum albumin (BSA; BovoStar) for 1 h for blocking and next incubated in phosphate-buffered saline (PBS) containing 0.5% BSA with a primary antibody solution. After that, the cells were incubated in PBS containing 0.5% BSA with the secondary antibodies conjugated to either Alexa Fluor 488 or rhodamine. Nuclei were stained with 4′,6-diamidino-2-phenylindole (DAPI; Biotium). LSM 510 Meta, LSM 700, or LSM 800 (Carl Zeiss) was used for visualization. Immunostained cell images were analyzed by Zeiss LSM Image Browser and Zen 2.1 (black; Carl Zeiss) and organized by Adobe Photoshop (Adobe) was used.

**Quantitation of cells containing aggresome**. In most cases, more than 100 cells were analyzed in each experiment of two (Fig. 2b,c; Supplementary Fig. 7b) or three (Fig. 7b; Supplementary Fig. 13b) biological replicates. For Fig. 4c, more than 50 cells were analyzed in each experiment. Two experienced independent investigators counted and rated the cells in a blinded way. For quantitation of cells containing aggresome or dispersed aggregates, cell images were analyzed by Zeiss LSM Image Browser (Carl Zeiss) and number of cells containing aggresome or dispersed aggregates was analyzed using Microsoft Excel (Microsoft).

**Western blotting**. Total-cell protein samples were separated by electrophoresis in an SDS-polyacrylamide gel, transferred to Hybond ECL nitrocellulose (Amersham), and probed with a primary antibody. After that, the following secondary antibodies were used to detect the primary antibody: a horseradish peroxidase (HRP)-conjugated goat α-mouse IgG antibody (cat. # AP124P, MilliporeSigma), HRP-conjugated goat α-rabbit IgG antibody (cat. # AP132P, MilliporeSigma), or HRP-conjugated rabbit α-goat IgG antibody (cat. # A5420, MilliporeSigma). Western blot images were obtained on a chemiluminescence imaging system (Amersham Imager 600, GE Healthcare). For organizing western blotting images, Adobe Photoshop (Adobe) was used. Signal intensities of western blot bands were quantitated in the ImageJ software (National Institutes of Health, Bethesda, MD).

**Immunoprecipitations**. Cell pellets were resuspended in NET-2 buffer [50 mM Tris-HCl (pH 7.4), 150 mM NaCl, 1 mM phenylmethylsulfonyl fluoride (PMSF; Sigma), 2 mM benzamidine hydrochloride (Sigma), 0.05% NP-40 (IGEPAL® CA-630; Sigma), 10 mM sodium fluoride (Sigma), and 0.25 mM sodium orthovanadate (Sigma)]. The resuspended cells were sonicated and centrifuged at $13,800 \times g$ for 10 min at 4 °C. For preclearing, protein G agarose 4B beads (Incospharm) were mixed with the cell extracts for 1 h at 4 °C. The precleared samples were incubated with various antibody-bound beads for 3 h at 4 °C. The beads were washed with NET-2 buffer five times, and the bead-bound proteins were eluted with 2× sample buffer [10% β-mercaptoethanol, 4% SDS, 100 mM Tris-HCl (pH 6.8), 15% glycerol, and 0.008% bromophenol blue].

**The in vitro GST pull-down assay**. This assay was carried out using recombinant GST, GST-CTIF, GST-eEF1A1, GST-DCTN1, and 6xHis-UPF1. Escherichia coli BL21(DE3)pLysS cells were transformed with the plasmid encoding either a GST-fusion protein or 6xHis-UPF1. The recombinant protein was induced by the addition of 0.5 mM isopropyl β-D-1-thiogalactopyranoside and additional incubation for 2 or 3 h. After that, the cells were harvested and resuspended in lysis buffer [50 mM Tris-HCl (pH 8.0), 150 mM NaCl, 0.5% Triton X-100, 1 mM dithiothreitol (DTT), 10% (v/v) glycerol, 2 mM benzamidine, and 1 mM PMSF] and sonicated. The E. coli lysate expressing a GST-fusion protein was mixed with the lysate expressing 6xHis-UPF1. The mixture was incubated in incubation buffer [10 mM HEPES (pH 7.4), 1.5 mM magnesium acetate, 150 mM potassium acetate, 2.5 mM DTT, and 0.05% NP-40] for 30 min at 4 °C. Then, the glutathione Sepharose 4B resin was added to the mixture and incubated for 2 h. The protein-bound resin was washed four times with incubation buffer. The resin-bound proteins were resolved by SDS-PAGE followed by western blotting.

**Line-scan confocal microscopy and image acquisition.** Multicolor video rate line-scan confocal microscopy was performed for single-particle tracking in live cells[43]. GFP-CTIF and SiR-conjugated tubulin (SiR-tubulin, Spirochrome, SC002) were excited by two lasers (Cobolt MLDTM 488 nm, 60 mW; Cobolt MLDTM 638 nm, 100 mW), respectively. The incident beams and emission from the fluorophores were reflected and transmitted by a dichroic beam splitter (Semrock, Di01-R405/488/561/635). Two galvano scanning mirrors (Cambridge Technology 6231H, 15 mm) scanned the field of view with the linear focus and reproduced the focal image in the camera. A high NA objective (Olympus UPlanSApo 100×, NA = 1.4, oil immersion) in an inverted microscope (Olympus IX51) focused the excitation lasers and gathered the emitted photons. The emitted light was filtered by wavelengths (Chroma, 59007m for SiR and Semrock FF01-520/35-25 for GFP) using a DV2 multichannel imaging system (Photometrics) that was placed in front of the electron-multiplying charged-couple device (EMCCD) camera (Andor iXon Ultra). All images were recorded by the SOLIS imaging software (Andor) every 100 ms, while the recording and scanning were synchronously controlled by a LabView script. To maintain the viability of cells during fluorescence imaging, the imaging was performed in a $CO_2$ incubation chamber at 37 °C (Live Cell Instrument). To colocalize the SiR-tubulin (microtubule) and GFP-CTIF particles, the image of fluorescent beads on the imaging plane was acquired as a reference for the mapping procedure.

**Analysis of GFP-CTIF aggregates migrating toward the aggresome.** The position of GFP-CTIF aggregates was determined with nanometer accuracy in DiaTrack 3.03 software[64]. The data were analyzed by means of Origin Pro 8 (OriginLab) and MATLAB script (2017a, Mathworks). The translocation rate of each spot was calculated from its trajectory with a continuous motion except for stalled segments. Trajectories moving less than or equal to 5 pixels (118 nm per pixel) for 100 ms (time resolution) were analyzed. To determine the frequency of GFP-CTIF aggregates moving into the aggresome, the first 150 frames were analyzed. From the signals, we subtracted background noise and filtered the data by means of the convolution function in ImageJ (National Institutes of Health). The distribution of the angle (θ) values was obtained from the trajectories with both the distance of greater than or equal to 9 pixels and lifetime longer than 5 points (500 ms). The histogram was produced by automatic splitting of the data range into bins of equal size in Origin Pro 8.

**The TUNEL assay.** The In Situ Cell Death Detection Kit (Roche) was utilized for the TUNEL assay. In brief, cells were fixed with 3.65–3.80% formaldehyde for 1 h. Endogenous peroxidase activity was inhibited by the addition of 3% hydrogen peroxide (Sigma). The cells were permeabilized with 0.1% Triton X-100. Then, DNA strand breaks were labeled with tetramethylrhodamine red. Nuclei were stained with DAPI. The cells were visualized under a Zeiss confocal microscope (LSM 510 Meta and LSM700, Carl Zeiss). For quantitation of apoptotic cells, more than 100 cells were analyzed for each experiment on three biological replicates.

**Reporting summary.** Further information on research design is available in the Nature Research Reporting Summary linked to this article.

## Data availability

The data supports the findings of this study are available within the article and its Supplementary information files or from the corresponding author upon reasonable request. The source data for Figs. 2b–c, 3a, 4a, 4c, 5a-f, 6e, 7b, and Supplementary Figs. 1a–d, 3b-d, 4a-c, 7b-d, 8a-d, 9a-f, 11a, and 13b-d are provided as a Source Data file. Source data are provided with this paper.

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

## Acknowledgements

We thank Drs. Akio Yamashita and Shigeo Ohno for providing the α-SMG6 antibody, Dr. Jens Lykke-Andersen for plasmid pCMV-Myc-DCP1A, and Dr. Ron R. Kopito for the plasmid expressing GFP-CFTR-ΔF508. This work was supported by a National Research Foundation (NRF) of Korea grant funded by the Korean government (Ministry of Science, ICT and Future Planning; NRF-2015R1A3A2033665 and NRF-2018R1A5A1024261 to Y.K.K.; and NRF-2017K1A1A2013241 to J.-B.L.) and by a Korea University Future Research grant to Y.K.K.; Y.P. was supported in part by the NRF funded by the Ministry of Education (NRF-2019R1A6A3A13094238).

## Author contributions

Y.P., J.P., H.J.H., and Y.K.K. conceived and designed the experiments. Y.P., J.P., H.J.H., and J.C. conducted the experiments. B.K. conducted the single-particle tracking experiments and analyzed the imaging data under the supervision of J.-B.L. and Y.K.K. K.J. conducted the GST pull-down experiments. Y.P., J.P., H.J.H., B.K., J.-B.L., and Y.K.K. wrote the manuscript.

## Competing interests

The authors declare no competing interests.
