## [Peer Review File · Nature Communications]

Reviewers' comments:

Reviewer #1 (Remarks to the Author):

The nonsense-mediated mRNA decay pathway is responsible for degrading mRNAs containing premature termination codons. Degradation of the mRNA prevents further production of truncated, potentially misfolded proteins; however, because NMD requires translation, some amount of aberrant protein is invariably produced. In this manuscript, Park and colleagues investigate a potential role for the NMD factor UPF1 in trafficking misfolded proteins to the aggresome for degradation. The authors present evidence that polypeptides produced from a model NMD substrate are localized to the aggresome and that UPF1 can also traffic to the aggresome under certain conditions. Cellular and in vitro interaction data suggest that this localization is mediated by UPF1 interactions with the CTIF-eEF1A1-DCTN1 complex. Additionally, the authors report that depletion of UPF1, but not other NMD pathway components, causes defective targeting of aggregates to the aggresome. A link between RNA quality control and protein quality control functions of UPF1 is interesting, but the manuscript does not support the conclusions drawn, particularly with regard to the role of UPF1 phosphorylation (see point 1). Further, the mechanisms of potential UPF1 involvement in aggregation and turnover of misfolded proteins are unclear.

Major points:

1. Throughout the manuscript, the authors use the UPF1 G495R;G497E mutant to make inferences about the importance of UPF1 hyperphosphorylation in aggresome targeting. In addition to being hyperphosphorylated, this mutant of UPF1 is deficient for ATP binding (see, for example, Franks et al., 2010 or Kurosaki et al., 2014). Because this mutant cannot bind ATP, it becomes locked on non-substrate RNAs, leading to its hyperphosphorylation (Kurosaki et al., 2014, Lee et al., 2015, and Durand et al., 2016). The authors do not even mention that the “hyperphosphorylated” UPF1 is also unable to bind ATP and is thus inactive in NMD, a major oversight. Further, because this UPF1 mutant abolishes ATPase and NMD activities of UPF1, it is impossible to conclusively attribute its effects on aggresome formation and trafficking to hyperphosphorylation. To establish whether ATPase activity or phosphorylation of UPF1 are required for the various cellular effects shown, many experiments should be repeated with ATPase-competent and ATPase-deficient UPF1 mutants lacking SMG1 phosphorylation sites. If hyperphosphorylation is indeed important for the UPF1 behaviors shown here, mutation of the phosphorylation sites should reverse the effects of the G495R;G497E mutation. Because there are many possible phosphorylation events, these experiments should be performed with mutants that completely remove potential phosphorylation sites (as in the Durand et al., paper) rather than the 4SA mutant used here.

2. Figure 3B, Figure 5, and Figure S5: The authors perform immunoprecipitation experiments in the presence of RNase A, but no controls are shown to demonstrate that the RNase treatment was effective in disrupting RNA-mediated protein interactions. Without this information, it is difficult to assess whether the observed interactions are truly direct. Similarly, because the GST pulldown experiments in

Figure S5B were performed in crude E. coli lysates, it is possible that the observed interactions were bridged by RNA.

3. Figure 4: The representative image of CFTR-deltaF508 localization upon UPF1 depletion and add-back of the wild-type protein looks very different from the “HP” rescue image, but the quantification of these two conditions is similar. From the images shown, it appears that the rescue with the wild-type protein was not effective, making it difficult to draw conclusions from this experiment.

4. Figure 5: Western blot quantification should include measures of error and statistical significance. Also, the procedures used to perform and quantify western blots should be included in the methods section.

5. Figure 6: The authors present evidence that knockdown of UPF1 impairs trafficking of GFP-CTIF to the aggresome. To interpret these findings, it is important to establish whether the MTOC and microtubule organization are intact in the cells treated with UPF1 siRNA. Without this information, it is premature to conclude that UPF1 has a specific role in promoting aggresome targeting.

6. Several statements in the text lack support from the data in this manuscript or the prior literature:

a. The first paragraph of the discussion claims that the authors have shown “selective degradation of NMD-polypeptides by the aggresome-autophagy pathway,” but this is not accurate. They do not show that NMD target-derived polypeptides are degraded, and (as stated later in the discussion), there is no evidence that this mechanism is selective for NMD-polypeptides.

b. Page 3, lines 56-57: I am not aware of any evidence that the EJC slows translation termination, and no papers are cited in support of this claim.

c. Page 13, lines 285-7: The statement that “UPF1 binds to misfolded polypeptides via the CED complex” is not supported by the evidence shown. It would be more appropriate to conclude that UPF1 association with misfolded polypeptides depends on or is promoted by the CED complex.

d. Page 14, line 323: The authors have not shown that the CED complex interaction with polypeptides is stabilized, merely that it is reduced at steady-state.

e. As discussed above, numerous instances in which behavior of the G495R;G497E is attributed to hyperphosphorylation are unsupported, due to the ATP binding and NMD defects of this protein.

Minor points:

1. Genes/proteins should be referred to according to their official symbols (e.g. UPF3B instead of UPF3X, etc.).

2. It would be helpful to clarify that many polypeptides produced from mRNAs degraded by NMD are not truncated.

Reviewer #2 (Remarks to the Author):

Park et al. provided evidence that short peptides produced during nonsense-mediated mRNA decay (NMD) are concentrated on the perinuclear inclusion (aggresome) by an NMD-processing factor UPF1. Using an NMD reporter, evidence was presented that peptides produced during NMD can be found at the aggresome upon proteasome inhibitor treatment (MG132). This concentration was inhibited by knockdown of UPF1 and its upstream kinase SMG1, but not other components of NMD. UPF1 and SMG1 (as well as ATM) knockdown also suppressed aggresome of CFTR-Del508F, which is not a substrate of NMD. A hyper-phosphorylated mutant of UPF1 (HP) is concentrated at the aggresome and can restore aggresome formation in UPF1 knockdown cells while phosphorylation-deficient UPF1-4SA mutant cannot (see comments). It was further shown that UPF1-HP had a much stronger interaction with the CED complex involved in aggresome formation than UPF1-4SA mutant did. These interactions, however, are not MG132 dependent. Overall, this report provides evidence and a mechanism that truncated peptides produced during NMD, if not degraded by the proteasome, can be concentrated to the aggresome by a UPF1 phosphorylation-dependent manner. This study is in agreement with a previous finding indicating that nascent truncated/misfolded peptides can trigger aggresome formation when their degradation is inhibited by proteasome inhibition. The current study provides a plausible mechanism to explain the previous finding. Overall, this is an interesting report that provides a further understanding of the origin of aggresomes. This report is suitable for Nature Communication following proper revision.

Specific Comments:

1. A FLAG-Gpx1 NMD reporter was used for the study. Although the data are quite convincing, it would be better to include a second reporter to substantiate the conclusion further

2. Fig 3C. Why would Myc-UPF1-WT behave differently from Myc-UPF1-HP (phosphorylated form) and did not concentrate on the aggresome in response to MG132?

- Fig 4B. The rescue experiments by UPF1-WT, UPF1-HP, and UPF1-4SA in UPF1 knockdown cells are not convincing. The expression of transfected UPF1 should be identified by immuno-staining to more accurately assess their effects on aggresome formation. The UPF1-WT, again, is not effective in rescuing aggresome formation. A detailed discussion of this observation should be included.

- Related to the previous comment, the inhibitory effect of SMG1 and ATM on aggresome formation (using mutant CFTR) would suggest that under MG132 treatment condition, UPF1 is phosphorylated to a level that can support aggresome formation. However, WT UPF1 is not phosphorylated at the detectable levels under basal or MG132 conditions. Additional experiments or a convincing explanation should be provided.

-It would also be more compelling to document the localization of endogenous UPF1 to the aggresome.

- How does UPF1 regulate CFTR-aggresome formation?

3. "Puromycin treatment results in premature translation termination and generates polypeptidyl-puro" (Line 142). In most studies, puromycin treatment typically does not lead to single juxtannuclear inclusion (aggresome); it induces multiple foci referred as aggresome-like (ALIS; see the Reference cited and Pankiv et al. JBC, 2007). The formation of these foci does not require proteasome inhibition. In Supplemental Fig S1C and S2A, polypeptidyl-puro treatment did not lead to a typical ALIS phenotype; instead, an MG132-dependent aggresome-like inclusion was presented. The authors should comment on this result.

-

Other issues:

1. There are too many abbreviated jargons used throughout the manuscript. The average reader will have a hard time to navigate through TC, CBC, CBPs, eRF, EJC, PTC, CED..... in one paragraph.

Reviewer #3 (Remarks to the Author):

Review: "Nonsense-mediated mRNA decay factor UPF1 promotes aggresome formation" by Park et al.

In the above paper, the authors investigate the role of UPF1 in regulating the quality of mRNA. The results are interesting. Focusing on the single particle tracking aspects of the paper, the paper would profit from a better description of what the authors are doing. For example, expanding briefly on line-scan confocal microscopy, so it is not easily confused with line-scanning confocal microscopy, a different technique (used in a paper I was reviewing earlier in the day).

Another example: Lines 314-316: "we defined the degree of directional motion of GFP-CTIF aggregates toward the aggresome as an angle (θ) between the direction of movement of the GFP-CTIF aggregates and the location of the aggresome (Fig. 6c)." The aggresome is a large object, so how do the authors define the "location of the aggresome?". In addition, the aggresome in the mUPF1 cell (Video S3) appears much larger than normal, taking up about a third of the nuclear periphery. How is it possible to calculate an angle with such an aggresome? A characterization of how the aggresome is affected under the different conditions is needed.

Lines 318-321: "These findings indicate that downregulation of UPF1, but not UPF2, significantly decreased the frequency of GFP-CTIF aggregates migrating toward the aggresome without affecting the translocation rate." Where does the frequency come into the play? According to the figure legend (Fig. 6d), there is a significant increase in the number of trajectories for the mUPF1 siRNA measurements, suggesting a total increase in particles. Combined with the abnormally large aggresome, the conclusion that the frequency of GFP-CTIF aggregates migrating towards the aggresome decreases is unfounded.

Why do the authors choose 5 pixels = 590 nm for the minimum movement for incorporation into the analysis? This seems to be a very small distance, the direction of which will be dominated by the

direction of the microtubules and not that of the overall transport. To substantiate their analysis, the authors should include other approaches. For example, what fraction GFP-CTIF in the field of view reach the aggresome during a fixed time interval? From the nice images, more information could be extracted from the single particle tracking data.

Minor comments

Figure S6: What is the velocity distribution for GFP-CTIF particles moving away from the aggresome? Is there a difference?

Lines 34, 102 and 304: single-molecule. These measurements are on single aggregates, but the signal is not coming from a single molecule. Please rephrase.

Responses to reviewer #1's comments

The nonsense-mediated mRNA decay pathway is responsible for degrading mRNAs containing premature termination codons. Degradation of the mRNA prevents further production of truncated, potentially misfolded proteins; however, because NMD requires translation, some amount of aberrant protein is invariably produced. In this manuscript, Park and colleagues investigate a potential role for the NMD factor UPF1 in trafficking misfolded proteins to the aggresome for degradation. The authors present evidence that polypeptides produced from a model NMD substrate are localized to the aggresome and that UPF1 can also traffic to the aggresome under certain conditions. Cellular and in vitro interaction data suggest that this localization is mediated by UPF1 interactions with the CTIF-eEF1A1-DCTN1 complex. Additionally, the authors report that depletion of UPF1, but not other NMD pathway components, causes defective targeting of aggregates to the aggresome. A link between RNA quality control and protein quality control functions of UPF1 is interesting, but the manuscript does not support the conclusions drawn, particularly with regard to the role of UPF1 phosphorylation (see point 1). Further, the mechanisms of potential UPF1 involvement in aggregation and turnover of misfolded proteins are unclear.

- **We appreciate the reviewer's interest in our work and the valuable comments. For details, please refer to our responses to the reviewers' comments below.**

Major points:

1. Throughout the manuscript, the authors use the UPF1 G495R;G497E mutant to make inferences about the importance of UPF1 hyperphosphorylation in aggresome targeting. In addition to being hyperphosphorylated, this mutant of UPF1 is deficient for ATP binding (see, for example, Franks et al., 2010 or Kurosaki et al., 2014). Because this mutant cannot bind ATP, it becomes locked on non-substrate RNAs, leading to its hyperphosphorylation (Kurosaki et al., 2014, Lee et al., 2015, and Durand et al., 2016). The authors do not even mention that the "hyperphosphorylated" UPF1 is also unable to bind ATP and is thus inactive in NMD, a major oversight. Further, because this UPF1 mutant abolishes ATPase and NMD activities of UPF1, it is impossible to conclusively

attribute its effects on aggresome formation and trafficking to hyperphosphorylation. To establish whether ATPase activity or phosphorylation of UPF1 are required for the various cellular effects shown, many experiments should be repeated with ATPase-competent and ATPase-deficient UPF1 mutants lacking SMG1 phosphorylation sites. If hyperphosphorylation is indeed important for the UPF1 behaviors shown here, mutation of the phosphorylation sites should reverse the effects of the G495R;G497E mutation. Because there are many possible phosphorylation events, these experiments should be performed with mutants that completely remove potential phosphorylation sites (as in the Durand et al., paper) rather than the 4SA mutant used here.

- **We apologize for the poor description of UPF1-G495R/G497E, which we renamed UPF1-HP in this study. In the original manuscript, we described the characteristics of this mutant as follows: “UPF1-HP contains two substitutions (G495R and G497E) within the helicase domain and becomes hyperphosphorylated because of inefficient dephosphorylation.” To present more detailed characteristics of this mutant, the sentence was updated as follows: “UPF1-HP contains two substitutions (G495R and G497E) within the helicase domain, and as a consequence, it lacks ATPase activity and becomes locked on mRNAs, leading to its hyperphosphorylation.”**
- **As suggested by the reviewer, we constructed an additional UPF1 variant: UPF1-HP-12A lacking all potential 12 phosphorylation sites (Durand et al. 2016) on the backbone of HP. Inefficient phosphorylation of UPF1-HP-12A, compared with UPF1-HP, was confirmed by Western blotting with an anti (α)-phospho-(S/T)Q antibody (Fig. 3a). Furthermore, we validated the function of UPF1-WT and its variant in NMD through complementation experiments. The results showed that NMD requires UPF1-WT, but not UPF1-HP, UPF1-HP-12A, or UPF1-HP-E3 mut. These new data were added into Supplementary Fig. 4 in the revised manuscript.**
- **UPF1-HP-12A exhibited inefficient aggresomal targeting under normal conditions or during MG132 treatment (Fig. 3b), suggesting that aggresomal targeting of UPF1 is dependent on its phosphorylation. Therefore, the original Fig. 3 was replaced with revised Fig. 3.**

- **Throughout the study, we repeated similar experiments using UPF1-HP-12A and replaced all the data involving UPF1-4SA with those from UPF1-HP-12A (see Figs. 4, 5a, 5c, and 7 and Supplementary Fig. 12d).**

2. Figure 3B, Figure 5, and Figure S5: The authors perform immunoprecipitation experiments in the presence of RNase A, but no controls are shown to demonstrate that the RNase treatment was effective in disrupting RNA-mediated protein interactions. Without this information, it is difficult to assess whether the observed interactions are truly direct. Similarly, because the GST pulldown experiments in Figure S5B were performed in crude *E. coli* lysates, it is possible that the observed interactions were bridged by RNA.

- **To address the reviewer's concern, we measured and compared the levels of endogenous *GAPDH* mRNAs in the extracts either treated or not treated with RNase A before IPs. After that, equal amounts of *in vitro*-synthesized firefly luciferase (FLuc) RNAs were added to the samples as a spike-in to adjust the data for differences among RNA preparations. The qRT-PCR results on total RNA in Fig. 3a (corresponding to the original Fig. 3b) show that the amount of endogenous *GAPDH* mRNA diminished almost to the basal level, confirming efficient removal of endogenous RNAs under our conditions. These new data were added in Supplementary Fig. 4a in the revised manuscript.**
- **In Fig. 3a (corresponding to original Fig. 3b), we did not test any coimmunopurified proteins except GAPDH. Therefore, we carried out further experiments to validate efficient RNA removal after RNase A treatment. To this end, we analyzed coimmunopurified CBP80, which is known to interact with UPF1 in an RNA-dependent manner (Hwang, et al., 2010; Kurosaki et al. 2014; Rufener et al., 2013), in the IP of FLAG-UPF1. Consistently with previous reports, we observed that RNase A treatment drastically reduced the amount of coimmunopurified CBP80 in the IP of UPF1, indicating that an RNA-mediated protein–protein interaction was almost completely prevented under our conditions. These new data were added into Supplementary Fig. 8c,d.**

3. Figure 4: The representative image of CFTR- Δ F508 localization upon UPF1 depletion and add-back of the wild-type protein looks very different from the “HP” rescue image, but the quantification of these two conditions is similar. From the images shown, it appears that the rescue with the wild-type protein was not effective, making it difficult to draw conclusions from this experiment.

- **Reviewer #2 also mentioned the same issue regarding complementation experiments (the second paragraph of comment #2 of reviewer #2). Furthermore, reviewer #2 suggested that “the expression of transfected UPF1 should be identified by immuno-staining to more accurately assess their effects on aggresome formation.” Furthermore, reviewer #1 asked us to use UPF1-HP-12A to clearly determine the phosphorylation effect (comment #1 of reviewer #1). Therefore, to address both reviewers’ comments, we repeated the same complementation experiments using UPF1-HP-12A and then measured the efficiency of aggresome formation in the cells expressing exogenous UPF1.**
- **The new data showed that the UPF1 downregulation triggered dispersion of the CFTR- Δ F508 aggresome. The dispersion pattern was partially or remarkably reversed by the expression of Myc-UPF-WT or Myc-UPF1-HP, respectively. On the other hand, the expression of Myc-UPF1-HP-12A failed to reverse the observed dispersion of CFTR- Δ F508, indicating that efficient aggresome formation involves UPF1 hyperphosphorylation. Therefore, all the data in Fig. 4 of the original manuscript were replaced by new Fig. 4.**
- **Our findings showed that Myc-UPF1-WT partially reversed the CFTR- Δ F508 aggresome dispersion caused by UPF1 downregulation, as compared with Myc-UPF1-HP. However, the same UPF1-WT almost completely restored NMD efficiency (Supplementary Fig. 4b,c). Therefore, the observed partial restoration of aggresome formation by Myc-UPF1-WT may be due to a local change in structure (important for aggresome formation) within UPF1 after N-terminal Myc tagging. Alternatively, exogenously expressed UPF1 may require additional time for the full ability to trigger efficient aggresome formation (e.g., its association with a misfolded polypeptide, its uncharacterized protein folding, formation of its complex with CED, or its**

movement [toward the aggresome] in the form of a complex with the misfolded polypeptide). The above description was inserted into the corresponding part of the Results section.

4. Figure 5: Western blot quantification should include measures of error and statistical significance. Also, the procedures used to perform and quantify western blots should be included in the methods section.

➤ **As suggested, all Western blot images were quantitated and summarized (see Supplementary Table 1) with standard deviations and statistical significance. The detailed description of Western blotting and image quantitation methods was also inserted into the Methods section.**

5. Figure 6: The authors present evidence that knockdown of UPF1 impairs trafficking of GFP-CTIF to the aggresome. To interpret these findings, it is important to establish whether the MTOC and microtubule organization are intact in the cells treated with UPF1 siRNA. Without this information, it is premature to conclude that UPF1 has a specific role in promoting aggresome targeting.

➤ **To ease the reviewer’s concern, we analyzed the effect of UPF1 downregulation on the MTOC and microtubule organization by monitoring the distributions of γ -tubulin and microtubules. Our results of confocal microscopy showed that UPF1 downregulation only marginally affected MTOC and microtubule organization. These new data were added in Supplementary Fig. 11c.**

6. Several statements in the text lack support from the data in this manuscript or the prior literature:

a. The first paragraph of the discussion claims that the authors have shown “selective degradation of NMD-polypeptides by the aggresome-autophagy pathway,” but this is not accurate. They do not show that NMD target-derived polypeptides are degraded, and (as stated later in the discussion), there is no evidence that this mechanism is selective for NMD-polypeptides.

➤ **We agree. Originally, we started our project with the investigation of UPF1-mediated aggresomal targeting of NMD-polypeptides. However, subsequent experiments indicated that other truncated and misfolded polypeptides as well as NMD-polypeptides are also delivered to the aggresome in an UPF1-dependent manner. Therefore, in light of these findings, we toned down and rewrote our description in the Discussion section as follows: “In this study, we unravel a previously unappreciated role of UPF1 in the specific targeting of misfolded polypeptides to the aggresome, where misfolded polypeptides will be eventually eliminated from the cells by the aggresome–autophagy pathway.” Accordingly, all other relevant sentences throughout the manuscript were properly revised too.**

b. Page 3, lines 56-57: I am not aware of any evidence that the EJC slows translation termination, and no papers are cited in support of this claim.

➤ **We agree with the reviewer. As requested, we revised the sentence as follows: “In accordance with conventional NMD, when newly synthesized mRNAs contain the exon junction complex (EJC) sufficiently downstream of the termination codon, they are recognized as NMD substrates. The cellular factors recruited to the termination codon—eukaryotic peptide chain release factors (eRF) 1 and 3, UPF1, and SMG1 (a kinase specific for UPF1)—form a SURF complex.”**

c. Page 13, lines 285-7: The statement that “UPF1 binds to misfolded polypeptides via the CED complex” is not supported by the evidence shown. It would be more appropriate to conclude that UPF1 association with misfolded polypeptides depends on or is promoted by the CED complex.

➤ **We agree. As suggested, the description was updated. Accordingly, the corresponding subtitle was also changed as follows: “Hyperphosphorylated UPF1 preferentially associates with misfolded polypeptides in a CED-dependent manner.”**

d. Page 14, line 323: The authors have not shown that the CED complex interaction with polypeptides is stabilized, merely that it is reduced at steady-state.

- **This comment is the same as a comment above (comment #6c). As requested, we deleted the terms such as “stabilization” throughout the manuscript, and the mentioned sentences were revised as follows: “these results suggest that UPF1 promotes aggresome formation of misfolded polypeptides at two distinct steps: it helps efficient formation of the CED complex containing misfolded polypeptides (Fig. 5) and ensures proper movement of the misfolded-polypeptide–associating CED complex to the aggresome (Fig. 6).”**

e. As discussed above, numerous instances in which behavior of the G495R;G497E is attributed to hyperphosphorylation are unsupported, due to the ATP binding and NMD defects of this protein.

- **As requested in comment #1, we repeated the same experiment using the UPF1-HP-12A mutant. Please refer to our responses to comment #1 for more details.**

Minor points:

1. Genes/proteins should be referred to according to their official symbols (e.g. UPF3B instead of UPF3X, etc.).

- **As suggested, UPF3X was changed to UPF3B throughout the manuscript. Accordingly, the figure labeling was changed too.**

2. It would be helpful to clarify that many polypeptides produced from mRNAs degraded by NMD are not truncated.

- **This is a valid suggestion. As proposed, we revised the relevant sentences for clarity throughout the manuscript. For instance, the sentence in the Abstract was changed as follows: “Because NMD necessitates a translation event to recognize a premature termination codon on mRNAs, truncated misfolded polypeptides (NMD-polypeptides) could potentially be generated from NMD substrates as byproducts.”**

Responses to reviewer #2's comments

Park et al. provided evidence that short peptides produced during nonsense-mediated mRNA decay (NMD) are concentrated on the perinuclear inclusion (aggresome) by an NMD-processing factor UPF1. Using an NMD reporter, evidence was presented that peptides produced during NMD can be found at the aggresome upon proteasome inhibitor treatment (MG132). This concentration was inhibited by knockdown of UPF1 and its upstream kinase SMG1, but not other components of NMD. UPF1 and SMG1 (as well as ATM) knockdown also suppressed aggresome of CFTR-Del508F, which is not a substrate of NMD. A hyper-phosphorylated mutant of UPF1 (HP) is concentrated at the aggresome and can restore aggresome formation in UPF1 knockdown cells while phosphorylation-deficient UPF1-4SA mutant cannot (see comments). It was further shown that UPF1-HP had a much stronger interaction with the CED complex involved in aggresome formation than UPF1-4SA mutant did. These interactions, however, are not MG132 dependent.

Overall, this report provides evidence and a mechanism that truncated peptides produced during NMD, if not degraded by the proteasome, can be concentrated to the aggresome by a UPF1 phosphorylation-dependent manner. This study is in agreement with a previous finding indicating that nascent truncated/misfolded peptides can trigger aggresome formation when their degradation is inhibited by proteasome inhibition. The current study provides a plausible mechanism to explain the previous finding. Overall, this is an interesting report that provides a further understanding of the origin of aggresomes. This report is suitable for Nature Communication following proper revision.

- **We appreciate the reviewer's perusal of our paper and the critical comments. For details, please see our responses below.**

Specific Comments:

1. A FLAG-Gpx1 NMD reporter was used for the study. Although the data are quite convincing, it would be better to include a second reporter to substantiate the conclusion further

- **We appreciate the reviewer's interesting suggestion. The main conclusion of our study is that UPF1 is involved in aggresomal targeting of various misfolded polypeptides including NMD-polypeptides, polypeptidyl-puro, and CFTR-ΔF508, as shown in Fig. 2 and Supplementary Fig. 3.**
- **At the start of our study, we employed FLAG-GPx1-Ter polypeptides as a reporter for monitoring misfolded polypeptides. After that, we used two different reporters for misfolded polypeptides: polypeptidyl-puro and CFTR-ΔF508. Therefore, in total, we employed three different reporters for monitoring misfolded polypeptides in this study. All the data indicate that UPF1 is engaged in aggresomal targeting of a variety of misfolded polypeptides rather than only NMD-polypeptides. Therefore, we made the relevant descriptions clearer throughout the text.**

2. Fig 3C. Why would Myc-UPF1-WT behave differently from Myc-UPF1-HP (phosphorylated form) and did not concentrate on the aggresome in response to MG132?

- **In case of UPF1-WT, it undergoes continuous alternation between phosphorylated and dephosphorylated states. At the steady-state level, only a small proportion of endogenous UPF1 molecules is known to be phosphorylated. Although a transient small proportion of UPF1 phosphorylation is sufficient for aggresomal targeting of misfolded polypeptides, it would be hard to detect this phosphorylation by Western blotting. On the other hand, most of UPF1-HP should remain hyperphosphorylated in the steady state. As a consequence, the UPF1-HP enriched in the aggresome may be easily detected under our conditions. In support of this notion, treatment of the cells with okadaic acid (a strong inhibitor of phosphatases) caused accumulation of hyperphosphorylated UPF1 and led to aggresomal enrichment of UPF1. In addition, we observed that greater amounts of CED components were enriched in the IP of UPF1 in the cells treated with okadaic acid. These supporting data were added into Supplementary Figs. 6 and 8b.**

- Fig 4B. The rescue experiments by UPF1-WT, UPF1-HP, and UPF1-4SA in UPF1 knockdown cells are not convincing. The expression of transfected UPF1 should be identified by immuno-staining to more accurately assess their effects on aggresome formation. The UPF1-WT, again, is not effective in rescuing aggresome formation. A detailed discussion of this observation should be included.

- **Reviewer #1 also brought up the same issue about complementation experiments (the second paragraph of comment #3 of reviewer #1). Furthermore, reviewer #1 asked us to use UPF1-HP-12A to clearly determine the phosphorylation effect. Therefore, to address both reviewers' comments, we repeated the same complementation experiments using UPF1-HP-12A and then measured the efficiency of aggresome formation in the cells expressing exogenous UPF1. Please refer to our responses to comment #3 of reviewer #1 for more details.**
- **The new data revealed that the UPF1 downregulation triggered dispersion of the CFTR-ΔF508 aggresome. The dispersion pattern was partially or remarkably reversed by the expression of Myc-UPF1-WT or Myc-UPF1-HP, respectively. In contrast, the expression of Myc-UPF1-HP-12A failed to reverse the observed dispersion of CFTR-ΔF508, indicating that efficient aggresome formation involves UPF1 hyperphosphorylation. Therefore, all the data in Fig. 4 of the original manuscript were replaced by these new data in the present manuscript.**
- **Our data showed that Myc-UPF1-WT partially reversed the dispersion of the CFTR-ΔF508 aggresome caused by UPF1 downregulation, as compared with Myc-UPF1-HP. However, the same UPF1-WT almost completely restored NMD efficiency (Supplementary Fig. 4b,c). Therefore, the observed partial restoration of aggresome formation by Myc-UPF1-WT may be due to a local change in structure (important for aggresome formation) within UPF1 after N-terminal Myc tagging. Alternatively, exogenously expressed UPF1 may require additional time for the full ability to trigger efficient aggresome formation (e.g., its association with a misfolded polypeptide, its uncharacterized protein folding, formation of its complex with CED, or its movement [toward the aggresome] in the form of a complex with the misfolded**

polypeptide). The above description was added into the corresponding subsection of the Results section.

- Related to the previous comment, the inhibitory effect of SMOG1 and ATM on aggresome formation (using mutant CFTR) would suggest that under MG132 treatment condition, UPF1 is phosphorylated to a level that can support aggresome formation. However, WT UPF1 is not phosphorylated at the detectable levels under basal or MG132 conditions. Additional experiments or a convincing explanation should be provided.

- **Although endogenous UPF1 is mostly unphosphorylated, it undergoes continuous alternation of phosphorylation and dephosphorylation. Transiently phosphorylated UPF1 may be sufficient for efficient aggresomal targeting of misfolded polypeptides. Indeed, as the reviewer mentioned, our data on *SMG1* siRNA or *ATM* siRNA indicate that during MG132 treatment, transient phosphorylation of UPF1 could support efficient aggresome formation, although we could not detect phosphorylated UPF1.**
- **To accumulate hyperphosphorylated UPF1, we treated the cells with okadaic acid (a strong inhibitor of phosphatases). The results revealed that treatment with okadaic acid caused accumulation of the hyperphosphorylated UPF1 and led to aggresomal enrichment of UPF1. These new data were inserted into Supplementary Fig. 6.**

-It would also be more compelling to document the localization of endogenous UPF1 to the aggresome.

- **Please refer to our responses to the above comment.**

- How does UPF1 regulate CFTR-aggresome formation?

- **We appreciate the reviewer's critical question. In case of NMD-polypeptides, UPF1 phosphorylated in the NMD pathway is engaged in aggresomal targeting of NMD-polypeptides. By contrast, in the case of misfolded polypeptides synthesized from non-NMD substrates, misfolded polypeptides may activate SMG1 or a stress-induced kinase (ATM), both of which can trigger UPF1**

phosphorylation. Indeed, we noted that downregulation of either SMG1 or ATM comparably inhibited the formation of the aggresome containing CFTR- Δ F508 (Supplementary Fig. 7). Therefore, considering that UPF1 promiscuously binds to mRNA with enrichment in the 3' untranslated region, it is possible that a neighboring UPF1 molecule may be phosphorylated via SMG1 or ATM, when misfolded polypeptides such as CFTR- Δ F508 are generated from mRNA. These clarifications are now present in the Discussion section (please see the third paragraph).

3. “Puromycin treatment results in premature translation termination and generates polypeptidyl-puro” (Line 142). In most studies, puromycin treatment typically does not lead to single juxtannuclear inclusion (aggresome); it induces multiple foci referred as aggresome-like (ALIS; see the Reference cited and Pankiv et al. JBC, 2007). The formation of these foci does not require proteasome inhibition. In Supplemental Fig S1C and S2A, polypeptidyl-puro treatment did not lead to a typical ALIS phenotype; instead, an MG132-dependent aggresome-like inclusion was presented. The authors should comment on this result.

- **During the revision phase, we characterized the properties of the aggresome containing polypeptidyl-puro. When the cells were treated with a lower concentration of puromycin, we observed a clear-cut aggresome, i.e., a single juxtannuclear inclusion. On the other hand, when the cells were treated with a higher concentration of puromycin, we detected both an aggresome and an additional aggresome-like structure (ALIS). Therefore, it seems that the formation of an aggresome or ALIS containing polypeptidyl-puro depends on puromycin concentration. These new observations were added into Supplementary Fig. 2b.**

Other issues:

1. There are too many abbreviated jargons used throughout the manuscript. The average reader will have a hard time to navigate through TC, CBC, CBPs, eRF, EJC, PTC, CED..... in one paragraph.

- **As suggested, we tried to minimize the use of abbreviations. Except for the TC and CBC, the above abbreviations are rather popular. Therefore, throughout the manuscript, “TC” and “CBC” were changed to “termination codon” and “nuclear cap-binding complex, respectively. Because CBP80 is a protein symbol, it is not considered an abbreviation.**

Responses to reviewer #3's comments

Review: "Nonsense-mediated mRNA decay factor UPF1 promotes aggresome formation" by Park et al.

In the above paper, the authors investigate the role of UPF1 in regulating the quality of mRNA. The results are interesting. Focusing on the single particle tracking aspects of the paper, the paper would profit from a better description of what the authors are doing. For example, expanding briefly on line-scan confocal microscopy, so it is not easily confused with line-scanning confocal microscopy, a different technique (used in a paper I was reviewing earlier in the day).

- **We appreciate the reviewer's meticulous examination of our work and the constructive criticism. For details, please refer to our responses to the comments below.**
- **We used two distinct confocal microscopes: a conventional confocal microscope in scanning mode and a line-scan confocal microscope designed for single-particle imaging. Line-scan confocal microscopy originates from video-rate line-scan confocal microscopy devised by Lee et al. (Biophysical J, 2012). Therefore, throughout the manuscript, we replaced "confocal microscopy" with "conventional confocal microscopy" to distinguish it from line-scan confocal microscopy for single-particle tracking.**

Another example: Lines 314-316: "we defined the degree of directional motion of GFP-CTIF aggregates toward the aggresome as an angle (θ) between the direction of movement of the GFP-CTIF aggregates and the location of the aggresome (Fig. 6c)." The aggresome is a large object, so how do the authors define the "location of the aggresome?". In addition, the aggresome in the mUPF1 cell (Video S3) appears much larger than normal, taking up about a third of the nuclear periphery. How is it possible to calculate an angle with such an aggresome? A characterization of how the aggresome is affected under the different conditions is needed.

- **The location of an aggresome was defined as the location of an aggresome periphery, which is enough to determine the degree of the directional motion**

of GFP-CTIF aggregates toward the aggresome. After defining the angle (θ), we needed only the information on the periphery of an aggresome. Therefore, for clarity, we changed “the location of the aggresome” to “the location of the aggresome periphery” in the Results section and figure legend.

- **A larger nuclear periphery in the video was unexpectedly generated while the movie was produced. Individual GFP-CTIF signals in the mUPF1-depleted cells (Video S3) were not as bright as those in undepleted or mUPF2-depleted cells. To better visualize GFP-CTIF movements, we enhanced the contrast of images of the mUPF1-depleted cells for Video S3, resulting in a much larger aggresome as seen in the movie. In fact, most aggresomes in mUPF1-depleted cells were much smaller, and their periphery was much sharper, enough to determine the location of the aggresome periphery (see Fig. 6b). Therefore, we added the above description to the legend of Supplementary Video 3.**

Lines 318-321: “These findings indicate that downregulation of UPF1, but not UPF2, significantly decreased the frequency of GFP-CTIF aggregates migrating toward the aggresome without affecting the translocation rate.” Where does the frequency come into the play? According to the figure legend (Fig. 6d), there is a significant increase in the number of trajectories for the mUPF1 siRNA measurements, suggesting a total increase in particles. Combined with the abnormally large aggresome, the conclusion that the frequency of GFP-CTIF aggregates migrating towards the aggresome decreases is unfounded.

- **We apologize for the inadequate description. The number of trajectories in Fig. 6d is simply the number of the trajectories analyzed for the study. In addition, we normalized the numbers of all trajectories to the total number of trajectories. Therefore, the number of trajectories in Fig. 6d does not reveal any significance of the migration frequency of GFP-CTIF.**
- **To determine the frequency, we counted the GFP-CTIF aggregates (per cell) that reached the aggresome during 15 s. The new data showed that mUPF1 downregulation, but not mUPF2 downregulation, significantly reduced the frequency of GFP-CTIF aggregates that reach the aggresome. These new data are now shown in Fig. 6e.**

Why do the authors choose 5 pixels = 590 nm for the minimum movement for incorporation into the analysis? This seems to be a very small distance, the direction of which will be dominated by the direction of the microtubules and not that of the overall transport. To substantiate their analysis, the authors should include other approaches. For example, what fraction GFP-CTIF in the field of view reach the aggresome during a fixed time interval? From the nice images, more information could be extracted from the single particle tracking data.

- **The minimum movement of 5 pixels was used only to present Fig. 6b so as to demonstrate how the trajectories look in the cells under different conditions. We have never used the constraint when analyzing the translocation rate of GFP-CTIF aggregates.**
- **As proposed by the reviewer, we also analyzed the number of GFP-CTIF aggregates that reach an aggresome during a fixed time interval (15 s) upon mUPF1 or mUPF2 downregulation. The findings revealed that UPF1 downregulation reduced the number of GFP-CTIF particles that reached the aggresome. These new data were inserted into Fig. 6e.**

Minor comments

Figure S6: What is the velocity distribution for GFP-CTIF particles moving away from the aggresome? Is there a difference?

- **As the reviewer requested, we analyzed GFP-CTIF particles moving away from the aggresome. The results revealed that the rate of the GFP-CTIF aggregates moving away from the aggresome was comparable to that toward the aggresome, suggesting that the movement of CTIF away from the aggresome—this step is necessary for efficient recycling of CTIF—is an active process rather than simple diffusion. These new data were added in Supplementary Fig 10b.**

Lines 34, 102 and 304: single-molecule. These measurements are on single aggregates, but the signal is not coming from a single molecule. Please rephrase.

- **As suggested, we changed “single-molecule” to “single-particle” throughout the manuscript.**

Reviewers' comments:

Reviewer #1 (Remarks to the Author):

The revised manuscript by Park et al is improved in several areas. In particular, the use of constructs in which 12 potential UPF1 phosphorylation sites are mutated is welcome. However, the biological significance and mechanisms of UPF1 involvement in trafficking of misfolded polypeptides remain unclear. I find that many of the experiments presented, while suggestive, are difficult to interpret. Please see below for detailed comments:

1. The mechanism by which UPF1 may affect aggresome formation continues to be unclear, and the proposed mechanism in Figure 7C seems incompatible with the data shown. The "HP" mutant used in this study is well known to be defective for NMD target selection, so the idea that the ATPase-deficient UPF1 recognizes a premature termination codon and then transports the misfolded nascent peptide to the aggresome is implausible.
2. Moreover, the significance of the morphological changes of the aggresome in UPF1 siRNA-treated cells is questionable. It appears as though the single aggresome punctum in normal cells is broadened into a more loosely organized collection of punctae in UPF1 knockdown cells. In Figure 3b, the HP-12A mutant seems to colocalize with the more poorly organized collection of aggresomal punctae, calling into question the importance of phosphorylation in targeting to aggresomal structures.
3. In the abstract, the authors claim that "the apoptosis induced by proteotoxic stresses depends on UPF1 hyperphosphorylation," but this interpretation is not consistent with the data shown in Figure 7b. Instead, apoptosis is induced when cells are treated with the proteasome inhibitor MG132 and UPF1 is depleted. This effect is partially rescued by expression of WT UPF1, the "HP" mutant, or the "HP" mutant lacking 12 potential SMG1 target residues. Perhaps the authors intended to say that UPF1 hyperphosphorylation suppresses apoptosis, as on page 17?
4. In response to point 4 in my original review, the authors say that they have provided detailed descriptions of the methods used for Western blot imaging and quantification, but no information is provided on imaging. Were the blots imaged with film or with methods that permit more accurate quantification? Also, I could not find any measures of statistical significance in Supplementary Table 1. It would be much better to have measures of error and statistical significance included with the western blot figures themselves, as is standard practice, rather than bury that information in a supplemental table.
5. In response to point 5 in my original review, in which I raised the possible disruption of MTOC and microtubule organization as a trivial explanation for the proposed role of UPF1 in trafficking GFP-CTIF to the aggresome, the authors provide Supplementary figure 11C, which they say shows that UPF1 downregulation "only marginally affected MTOC and microtubule organization." It is difficult to judge the small number of cells shown in Figure S11C, but it appears as though UPF1 knockdown causes the MTOC to change shape to a more elongated structure, reminiscent of the broadening of the aggresome into a more poorly organized structure observed in several of the main text figures. It appears possible, therefore, that UPF1 knockdown does indeed disrupt MTOC structure in such a way that might cause a

corresponding change in aggresome structure. Without more clearly resolving this question, it is very difficult to interpret the trafficking data shown.

6. Figure 5C. This figure shows co-purification of puromycylated peptides with UPF1 mutants. Curiously, the UPF1-HP-12A protein co-purifies with many puromycylated species, inconsistent with the authors' conclusion that this protein did not preferentially copurify with polypeptidyl-puro. It is true that the pattern of species that co-purify with this mutant is different than those co-purified with the HP and HP-E3 mutants, but it still associates with a significant quantity of such nascent peptides.

Reviewer #2 (Remarks to the Author):

The primary concerns from the previous review are the lack of a definitive biochemical proof that UPF1 phosphorylation is involved in aggresome formation and how CFTR-Del508F, which is not an NMD substrate, becomes engaged with the SMG1/ATM-UPF1 circuit. The revision has mostly addressed the first concern but not the second one. The authors have suggested that "misfolded polypeptides may activate SMG1 or a stress-induced kinase (ATM)". Accordingly, the authors should determine whether CFTR-Del508F mutant expression would affect SMG1/ATM activation and UPF1 phosphorylation. This information would significantly strengthen the manuscript.

Reviewer #3 (Remarks to the Author):

The authors have adequately addressed my comments.

Responses to reviewer #1's comments

The revised manuscript by Park et al is improved in several areas. In particular, the use of constructs in which 12 potential UPF1 phosphorylation sites are mutated is welcome. However, the biological significance and mechanisms of UPF1 involvement in trafficking of misfolded polypeptides remain unclear. I find that many of the experiments presented, while suggestive, are difficult to interpret. Please see below for detailed comments:

➤ **We appreciate the reviewer's time and effort to evaluate our work.**

1. The mechanism by which UPF1 may affect aggresome formation continues to be unclear, and the proposed mechanism in Figure 7C seems incompatible with the data shown. The "HP" mutant used in this study is well known to be defective for NMD target selection, so the idea that the ATPase-deficient UPF1 recognizes a premature termination codon and then transports the misfolded nascent peptide to the aggresome is implausible.

➤ **We agree with the reviewer. Our main conclusion is that UPF1 is involved in aggresomal targeting of various misfolded polypeptides including NMD-polypeptides, polypeptidyl-puro, and CFTR-ΔF508, as shown in Fig. 2 and Supplementary Fig. 3. As the reviewer pointed out, it seems that the current model is confusing. Therefore, we deleted the NMD pathway and instead highlighted the role of UPF1 in aggresome formation. Thus, the model was revised (Fig. 7c).**

2. Moreover, the significance of the morphological changes of the aggresome in UPF1 siRNA-treated cells is questionable. It appears as though the single aggresome punctum in normal cells is broadened into a more loosely organized collection of punctae in UPF1 knockdown cells. In Figure 3b, the HP-12A mutant seems to colocalize with the more poorly organized collection of aggresomal punctae, calling into question the importance of phosphorylation in targeting to aggresomal structures.

- **By definition, an aggresome is a large single misfolded-polypeptide-containing inclusion that is formed via accumulation around the centrosome. As noted by the reviewer, UPF1 downregulation caused the dispersion of the aggresome, in part generating a loosely organized collection of punctae or small aggregates. Although we do not know the exact step affected by UPF1, it is clear that UPF1 downregulation blocks the condensation and transport of misfolded polypeptides toward the centrosome.**
- **To clearly address the above phenomenon, we conducted a single-particle experiment to track the movement of aggregates toward the aggresome reported in the original study (before revisions). As presented in Fig. 6, the single-particle experiments showed that UPF1 indeed affects the migration of aggregates toward the aggresome.**

3. In the abstract, the authors claim that “the apoptosis induced by proteotoxic stresses depends on UPF1 hyperphosphorylation,” but this interpretation is not consistent with the data shown in Figure 7b. Instead, apoptosis is induced when cells are treated with the proteasome inhibitor MG132 and UPF1 is depleted. This effect is partially rescued by expression of WT UPF1, the “HP” mutant, or the “HP” mutant lacking 12 potential SMG1 target residues. Perhaps the authors intended to say that UPF1 hyperphosphorylation suppresses apoptosis, as on page 17?

- **As suggested, the text was revised as follows: “the apoptosis induced by proteotoxic stresses is suppressed by UPF1 hyperphosphorylation.”**

4. In response to point 4 in my original review, the authors say that they have provided detailed descriptions of the methods used for Western blot imaging and quantification, but no information is provided on imaging. Were the blots imaged with film or with methods that permit more accurate quantification? Also, I could not find any measures of statistical significance in Supplementary Table 1. It would be much better to have measures of error and statistical significance included with the western blot figures themselves, as is standard practice, rather than bury that information in a supplemental table.

- **We apologize for not showing statistical significance. As suggested, revised Fig. 5b,c now contains all quantification data with standard deviations and statistical significance. Accordingly, all Western blotting images in supplementary figures were quantitated too and are now presented with standard deviations and statistical significance (Supplementary Figs. 8 and 9).**
- **All Western blot images were obtained on a chemiluminescence imaging system (Amersham Imager 600, GE Healthcare), and signal intensities of the western blot bands were quantitated using the ImageJ software (version 1.42q, National Institutes of Health, Bethesda, MD). All these descriptions were added into the Methods section and Figure legend in the latest version of the manuscript.**

5. In response to point 5 in my original review, in which I raised the possible disruption of MTOC and microtubule organization as a trivial explanation for the proposed role of UPF1 in trafficking GFP-CTIF to the aggresome, the authors provide Supplementary figure 11C, which they say shows that UPF1 downregulation “only marginally affected MTOC and microtubule organization.” It is difficult to judge the small number of cells shown in Figure S11C, but it appears as though UPF1 knockdown causes the MTOC to change shape to a more elongated structure, reminiscent of the broadening of the aggresome into a more poorly organized structure observed in several of the main text figures. It appears possible, therefore, that UPF1 knockdown does indeed disrupt MTOC structure in such a way that might cause a corresponding change in aggresome structure. Without more clearly resolving this question, it is very difficult to interpret the trafficking data shown.

- **The confocal images in Supplementary Fig. 11c (mUPF1 siRNA and mUPF2 siRNA) by chance showed duplicated MTOCs rather than an elongated MTOC. To ease the reviewer’s concern, we repeated the same experiments using MEFs and HeLa cells. To present a greater number of the cells, we captured confocal images at lower magnification. The new data support our previous conclusion that UPF1 downregulation only marginally affects the MTOC and microtubule organization. Therefore, original Supplementary Fig.**

11c was removed, and the new images were inserted into Supplementary Fig. 12.

6. Figure 5C. This figure shows co-purification of puromycylated peptides with UPF1 mutants. Curiously, the UPF1-HP-12A protein co-purifies with many puromycylated species, inconsistent with the authors' conclusion that this protein did not preferentially copurify with polypeptidyl-puro. It is true that the pattern of species that co-purify with this mutant is different than those co-purified with the HP and HP-E3 mutants, but it still associates with a significant quantity of such nascent peptides.

➤ **We agree with the reviewer. However, we believe that this result makes sense because UPF1-HP-12A is still phosphorylated to some extent as compared with UPF1-WT (Fig. 3a). It is possible that a minimally phosphorylated UPF1-HP-12A may weakly associate with misfolded polypeptides showing a different binding pattern. Therefore, the text was changed as follows: “The IP data showed that a basal and small amount of newly synthesized (and potentially misfolded) polypeptidyl-puro coimmunopurified with UPF1-WT and UPF1-HP-12A (which was confirmed to be minimally phosphorylated; Fig. 3a), respectively (Fig. 5d). On the other hand, a greater amount of polypeptidyl-puro coimmunopurified with UPF1-HP and UPF1-HP-E3 mut, in contrast to UPF1-WT and UPF1-HP-12A.”**

Responses to reviewer #2's comments

The primary concerns from the previous review are the lack of a definitive biochemical proof that UPF1 phosphorylation is involved in aggresome formation and how CFTR-Del508F, which is not an NMD substrate, becomes engaged with the SMG1/ATM-UPF1 circuit. The revision has mostly addressed the first concern but not the second one. The authors have suggested that “misfolded polypeptides may activate SMG1 or a stress-induced kinase (ATM)”. Accordingly, the authors should determine whether CFTR-Del508F mutant expression would affect SMG1/ATM activation and UPF1 phosphorylation. This information would significantly strengthen the manuscript.

- **We are grateful for the reviewer's time and thorough review of our work.**
- **As requested, we carried out the suggested experiments. The results revealed that overexpression of GFP-CFTR-ΔF508 triggers UPF1 phosphorylation. The increased UPF1 phosphorylation was reversed by downregulation of either SMG1 or ATM. All these data suggest that accumulation of misfolded polypeptides triggers UPF1 phosphorylation via SMG1 and/or ATM. These new data were added in Supplementary Fig. 7d in the latest version of the manuscript. The corresponding parts of the Results (page 11) and Discussion (page 19) sections were revised accordingly.**

Responses to reviewer #3's comments

The authors have adequately addressed my comments.

- **We appreciate the reviewer's positive evaluation.**

REVIEWERS' COMMENTS:

Reviewer #1 (Remarks to the Author):

I continue to have reservations about the biological significance of the observations, but the authors have largely adequately addressed my prior requests for additional controls and clarification.

One point remains that can be addressed with minor edits:

From my prior point 6, I still don't think that the description of the polypeptidyl-puro co-IP experiments is accurate. I don't understand how the amount of polypeptidyl-puro that co-purified with UPF1-HP-12A can fairly be characterized as "basal". The revised text still gives the impression that the amounts of polypeptidyl-puro associated with UPF1-WT and UPF1-HP-12A are similar, which is not the case.

Reviewer #2 (Remarks to the Author):

The revision has addressed all main concerns.

Responses to reviewer #1's comments

I continue to have reservations about the biological significance of the observations, but the authors have largely adequately addressed my prior requests for additional controls and clarification.

- **We are grateful for the reviewer's time and thorough re-review of our work.**

One point remains that can be addressed with minor edits:

From my prior point 6, I still don't think that the description of the polypeptidyl-puro co-IP experiments is accurate. I don't understand how the amount of polypeptidyl-puro that co-purified with UPF1-HP-12A can fairly be characterized as "basal". The revised text still gives the impression that the amounts of polypeptidyl-puro associated with UPF1-WT and UPF1-HP-12A are similar, which is not the case.

- **To ease the reviewer's concern, we modified the description, as suggested by the reviewer as follows: "The IP data showed that, in agreement with our finding that UPF1-HP-12A is still phosphorylated to some extent as compared with UPF1-WT (Fig. 3a), slightly more amount of newly synthesized (and potentially misfolded) polypeptidyl-puro coimmunopurified with UPF1-HP-12A compared with UPF1-WT (Fig. 5d)".**

Responses to reviewer #2's comments

The revision has addressed all main concerns.

- **We appreciate the reviewer's positive evaluation.**